# A CRISPR Interference Screen of Essential Genes Reveals that Proteasome Regulation Dictates Acetic Acid Tolerance in *Saccharomyces cerevisiae*

Vaskar Mukherjee,[a] Ulrika Lind,[b] Robert P. St. Onge,[c,d] Anders Blomberg,[b] Yvonne Nygård[a]

[a]Department of Biology and Biological Engineering, Division of Industrial Biotechnology, Chalmers University of Technology, Gothenburg, Sweden
[b]Department of Chemistry and Molecular Biology, University of Gothenburg, Gothenburg, Sweden
[c]Stanford Genome Technology Center, Stanford University, Palo Alto, California, USA
[d]Department of Biochemistry, Stanford University, Palo Alto, California, USA

**ABSTRACT** CRISPR interference (CRISPRi) is a powerful tool to study cellular physiology under different growth conditions, and this technology provides a means for screening changed expression of essential genes. In this study, a *Saccharomyces cerevisiae* CRISPRi library was screened for growth in medium supplemented with acetic acid. Acetic acid is a growth inhibitor challenging the use of yeast for the industrial conversion of lignocellulosic biomasses. Tolerance to acetic acid that is released during biomass hydrolysis is crucial for cell factories to be used in biorefineries. The CRISPRi library screened consists of >9,000 strains, where >98% of all essential and respiratory growth-essential genes were targeted with multiple guide RNAs (gRNAs). The screen was performed using the high-throughput, high-resolution Scan-o-matic platform, where each strain is analyzed separately. Our study identified that CRISPRi targeting of genes involved in vesicle formation or organelle transport processes led to severe growth inhibition during acetic acid stress, emphasizing the importance of these intracellular membrane structures in maintaining cell vitality. In contrast, strains in which genes encoding subunits of the 19S regulatory particle of the 26S proteasome were downregulated had increased tolerance to acetic acid, which we hypothesize is due to ATP salvage through an increased abundance of the 20S core particle that performs ATP-independent protein degradation. This is the first study where high-resolution CRISPRi library screening paves the way to understanding and bioengineering the robustness of yeast against acetic acid stress.

**IMPORTANCE** Acetic acid is inhibitory to the growth of the yeast *Saccharomyces cerevisiae*, causing ATP starvation and oxidative stress, which leads to the suboptimal production of fuels and chemicals from lignocellulosic biomass. In this study, where each strain of a CRISPRi library was characterized individually, many essential and respiratory growth-essential genes that regulate tolerance to acetic acid were identified, providing a new understanding of the stress response of yeast and new targets for the bioengineering of industrial yeast. Our findings on the fine-tuning of the expression of proteasomal genes leading to increased tolerance to acetic acid suggest that this could be a novel strategy for increasing stress tolerance, leading to improved strains for the production of biobased chemicals.

**KEYWORDS** CRISPR interference, yeast, high-throughput screening, acetic acid tolerance, essential genes, transcriptional regulation, phenomics, proteasome, oxidative stress

Address correspondence to Yvonne Nygård, yvonne.nygard@chalmers.se.

Systematic profiling of relationships between genotypes and phenotypes provides a novel understanding of fundamental biology and suggests leads for improving strains for various biotechnology applications. Quantitative phenotyping of different

collections of strains with systematic genetic perturbations, such as the yeast deletion collection (1), the yeast green fluorescent protein (GFP) clone collection (2), or yeast over-expression collections (3, 4), has allowed the construction of yeast regulatory network models. Nonetheless, the functions of a large number of genes remain unknown, and many known genes may have more functions yet to be discovered. Notably, even small perturbations in the expression of genes can lead to large phenotypic changes (5).

In recent years, CRISPR interference (CRISPRi) technology has been demonstrated as a very efficient tool to alter gene regulation (6). This technology exploits RNA-guided, endonuclease-dead Cas9 (dCas9), or other CRISPR-associated proteins, for the controlled downregulation of genes by directing dCas9 fusions to their promoter region (7). This allows alteration of the expression of essential genes, as partial loss-of-function phenotypes can be induced by the conditional expression of *dCas9* and the target-gene-specific guide RNA (gRNA). Furthermore, as the strength of the expression alteration is greatly dependent on the efficiency and positioning of the gRNA, one can study a gradient of repression by testing multiple gRNA sequences for each target gene (8, 9). Based on this technology, several CRISPRi strain libraries were constructed for many species, including *Saccharomyces cerevisiae* (9–13).

In the first CRISPRi library constructed for yeast (12), transcriptional interference was achieved with an integrated dCas9-Mxi1 repressor (14) and a tetracycline-regulatable repressor (TetR) that controls the expression of the gRNA (8). In this strain collection of roughly 9,000 strains, nearly 99% of the essential and 98% of the respiratory growth-essential genes have been targeted with up to 17 gRNAs per target gene (12). Recently, the construction and phenotypic screening of CRISPR technology-based *S. cerevisiae* libraries have been demonstrated to be very efficient to identify bioengineering genetic candidates to increase the production of $\beta$-carotene or endoglucanase (15), regulate polyketide synthesis (16), or improve tolerance to furfural (11) or lignocellulose hydrolysates (13).

Lignocellulose hydrolysates contain not only fermentable sugars but also various amounts of other compounds, including furfural, different weak acids, and phenolic compounds, that inhibit yeast growth (reviewed in reference 17). Among these compounds, toxicity by acetic acid is one of the most limiting factors for the production of alternative fuels and chemicals from lignocellulosic biomass using *S. cerevisiae*. Acetic acid is formed during the hydrolysis of biomass and is inhibitory to yeast (17), and lignocellulosic hydrolysates may contain 1 to 15 g/liter (17 to 250 mM) acetic acid, depending on the feedstock and pretreatment methods employed (reviewed in reference 18). Tolerance to acetic acid is a very complex trait, where many genetic elements together control the phenotype (19). As a result, rationally designing acetic acid-tolerant strains is challenging (18).

In this study, a CRISPRi library (12) was used to screen essential and respiratory growth-essential genes for roles in providing tolerance to acetic acid in *S. cerevisiae*. The library was characterized using the automated high-resolution and high-throughput Scan-o-matic platform (20), where each strain is analyzed separately for its growth rate on solid medium. A set of strains with interesting acetic acid growth profiles was verified in liquid medium, and the repression of some of these genes was verified by quantitative PCR (qPCR). The library enabled us to confirm previously known genes involved in the response to acetic acid and to identify several novel genes, the regulation of which could be altered to increase tolerance to acetic acid and thereby improve production hosts for the production of biocommodities from lignocellulosic biomass.

## RESULTS

**High-throughput phenomics of the CRISPRi strains.** To identify genes involved in the tolerance of *S. cerevisiae* to acetic acid, we performed a high-throughput growth screen of a CRISPRi library (9,078 strains) targeting essential and respiratory growth-essential genes with an integrated dCas9-Mxi1 repressor (12). Growth phenotyping of the CRISPRi library was performed using the Scan-o-matic system, providing high-

**CRISPRi library**

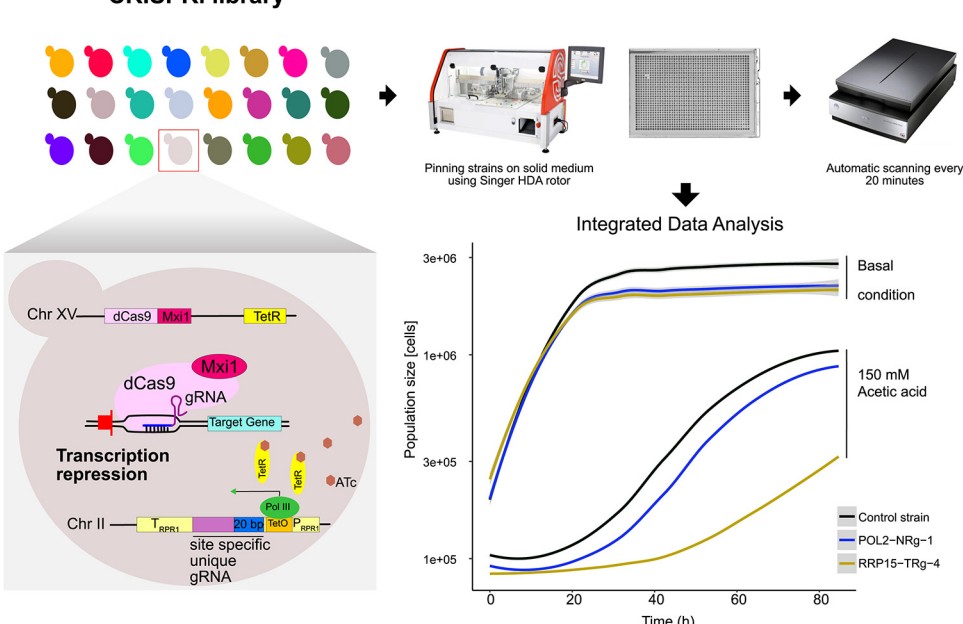

FIG 1 Constitutively expressed dCas9-Mxi1 and the tetracycline-regulatable gRNA expression system induce transcription repression of essential or respiratory growth-essential genes. Each strain in the library was phenotyped individually for growth on solid medium with 150 mM acetic acid or in basal medium lacking acetic acid using the Scan-o-matic platform.

resolution growth curves on solid medium for each strain (20) (Fig. 1). The potential bias between plates and runs was minimized via spatial normalization over the plates. The screens were independently duplicated, resulting in >27,000 images in total, and the image analysis generated >42 million data points and >140,000 growth curves. Our large-scale screen showed rather good repeatability (Fig. 2A). Linear regression, taking all strains into account, showed that 22% (coefficient of determination, i.e., $R^2$, of 0.22; $F$ test $P$ value of <2.2e−16) of the phenotypic variability between the two independent screens could be explained by the linear model (Pearson correlation coefficient [$r$] of 0.47). However, taking only the strains with distinct phenotypes into account, i.e., statistically significant acetic acid-sensitive or -tolerant strains (Fig. 2A), 79% ($R^2 = 0.79$; $F$ test $P$ value of <2.2e−16) of the phenotypic variability between the two independent experiments could be explained by the linear model ($r = 0.89$).

The CRISPRi strains showed limited phenotypic effects under basal conditions, and the generation time (GT) of 8,958 strains (99% of the strains of the library) was within ±10% of the generation time of the control strain (Fig. 2B). Only 92 strains (1%) displayed complete growth inhibition under basal conditions.

**CRISPRi-based gene repression imposed large phenotypic effects under acetic acid stress.** In contrast to basal medium, large variations in generation times were observed among the CRISPRi strains at 150 mM acetic acid (AA$_{150 mM}$) (Fig. 2B and C). A great proportion of the CRISPRi strains displayed slower growth in response to acetic acid, with 1,040 strains (~11%) having >10% higher generation times than the control strain. It was also clear from the growth curves that strains in acetic acid medium exhibited a rather long lag phase before growth resumed (Fig. 1). Still, 133 strains (~1%) displayed a >10% shorter generation time than the control strain in response to acetic acid (Fig. 2B). In conclusion, the addition of acetic acid to the growth medium had a great impact on the growth of many of the strains in the CRISPRi library. The raw data and all the subsequent analytical outputs for all strains in the library are available in the GitHub repository (https://github.com/mukherjeevaskar267/CRISPRi_Screening _AceticAcid/blob/main/BIG_DATA).

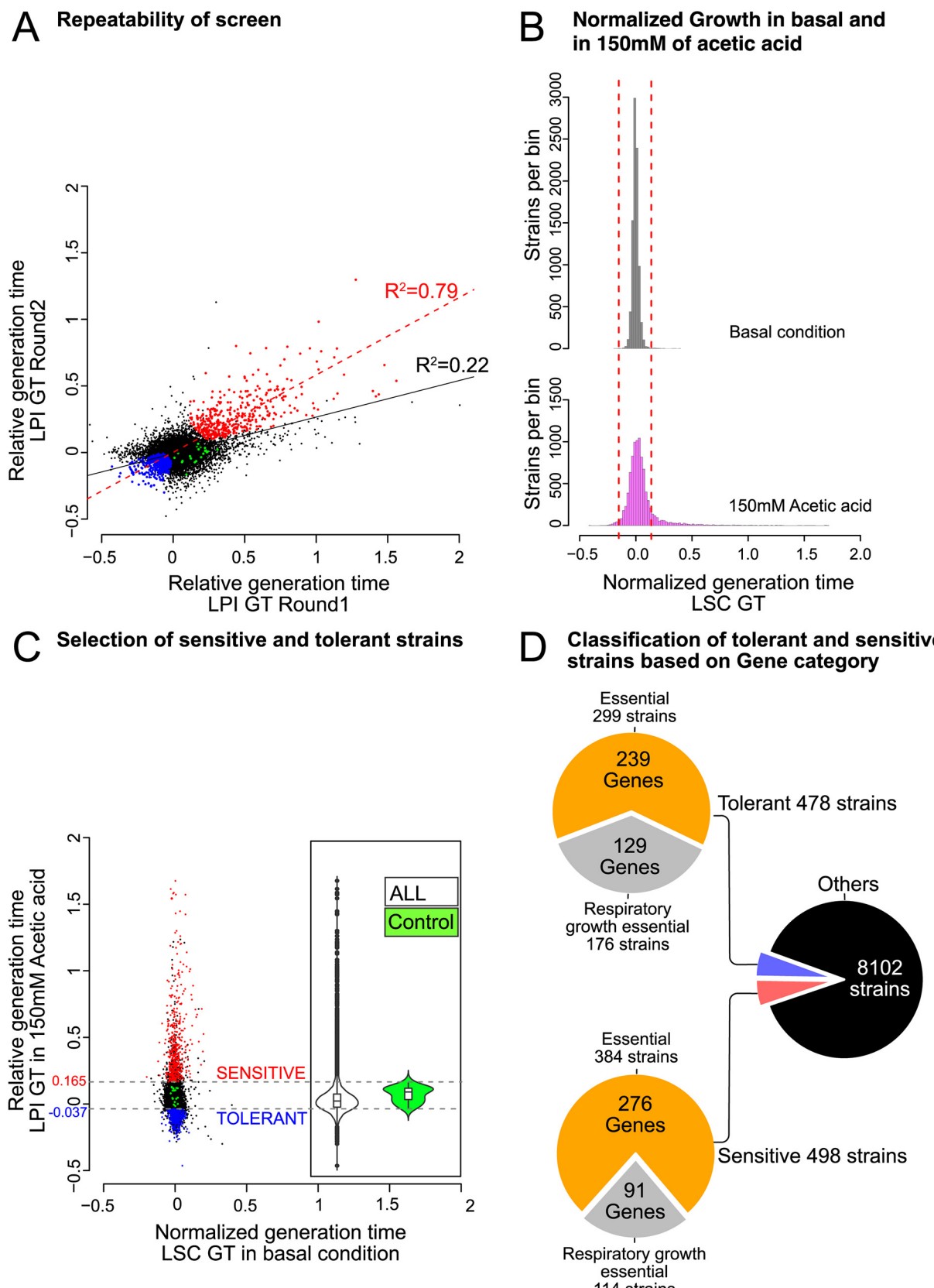

**FIG 2** The CRISPRi strains showed minor phenotypic variation under basal conditions and large phenotypic variation under acetic acid stress. (A) Scatterplot displaying the reproducibility of the two Scan-o-matic screenings. The means from the three LPI GT replicates for each strain are plotted, with control strains in green, acetic acid-sensitive strains in red, acetic acid-tolerant strains in blue, and the remaining

**Integrative data analysis connected yeast essential genes to acetic acid tolerance and sensitivity.** In order to study gene-specific effects on acetic acid tolerance/sensitivity, we constructed relative generation times (LPI [log phenotypic index]) where growth in acetic acid was put in relation to growth in basal medium. Thus, those strains that exhibited a general growth defect and grew poorly in both media were not identified as specifically sensitive to acetic acid.

A combined statistical (false discovery rate [FDR]-adjusted $P$ value of ≤0.1) and effect size threshold was applied, which allowed the identification of 959 strains (corresponding to 665 genes) as acetic acid sensitive or tolerant (Fig. 2D). Out of these, 478 strains with gRNAs targeting a total of 370 genes had significantly decreased relative generation times (Fig. 2D) (see https://github.com/mukherjeevaskar267/CRISPRi_Screening_AceticAcid/blob/main/BIG_DATA/Data3.xlsx) and thereby displayed acetic acid tolerance. The decrease in the relative generation time seen was relatively small, with only a few strains showing a higher level of improvement and with RPN9-TRg-4 (targeting *RPN9*, encoding a regulatory subunit of the 26S proteasome) (27% improvement) and RGL1-NRg-7 (targeting *RGL1*, encoding a regulator of Rho1p signaling) (18% improvement) being the most acetic acid-tolerant strains identified. A total of 498 strains, with gRNAs targeting a total of 367 genes, displayed acetic acid sensitivity (Fig. 2D) (see https://github.com/mukherjeevaskar267/CRISPRi_Screening_AceticAcid/blob/main/BIG_DATA/Data4.xlsx). Out of these, 17 strains that grew well under basal conditions were completely inhibited (or the strains grew extremely slowly, with a generation time of >48 h) in the presence of 150 mM acetic acid. The range of sensitivities was rather wide, and the relative generation time for 34 strains was >2-fold compared to that of the control strain, with ARC40-NRg-3 (targeting *ARC40*, encoding a subunit of the ARP2/3 complex) (219% extension) and VPS45-NRg-4 (targeting *VPS45*, encoding a protein essential for vacuolar protein sorting) (206% extension) being the most acetic acid-sensitive strains. Thus, a rather large number of CRISPRi strains showed an altered response to acetic acid, where about half showed increased sensitivity and half showed increased tolerance.

**Growth in liquid medium and qPCR expression analysis validated the large-scale phenomics results.** To validate the phenomics data obtained from cultures grown on solid medium, the growth of 183 strains (including sensitive and tolerant strains as well as some controls) was also analyzed in liquid medium. In the liquid validation experiment, both 150 mM and 125 mM acetic acid media were included, as the phenotypic effects were seen to be more drastic in liquid than on solid medium. A high proportion of the strains did not grow at all in liquid medium at 150 mM, the concentration that was used for the screen on solid medium. The relative generation time in liquid medium with 125 mM of acetic acid showed a strong correlation ($r = 0.86$) with the corresponding Scan-o-matic data for growth on solid medium (Fig. 3; for representative growth curves of selected strains in liquid medium, see Fig. S1A in the supplemental material). Linear regression showed that 73% ($R^2 = 0.73$; $F$ test $P$ value of <2.2e−16) of the phenotypic variation between these two independent experimental methods can be explained by the linear model. It should be noted that some strains can display, for biological reasons, different growth responses on solid and liquid media (20). We concluded that the data from the large-scale screen on solid medium was in excellent agreement with the data from the liquid growth analysis.

**FIG 2** Legend (Continued)

strains in black. The linear regression of the data is displayed with a black line for all strains and with a red line for the acetic acid-sensitive and -tolerant strains. (B) Histogram of the normalized generation times for each CRISPRi strain under basal conditions (gray) and with 150 mM acetic acid (magenta). Strains outside the two red dashed lines have generation times that are 10% shorter or 10% longer than that of the control strain. (C) Scatterplot showing the normalized generation time for each CRISPRi strain under basal conditions and the relative generation time in medium with 150 mM acetic acid. Each point indicates the mean for all the replicates ($n = 6$ [when some of the replicates failed to grow, $n = 3$ to 6]). The data for the CRISPRi control strains are indicated in green, those for the acetic acid-sensitive strains are in red, those for acetic acid-tolerant strains are in blue, and those for all other strains are in black. The LPI GT threshold is indicated with a gray dashed line. (Inset) Violin plots displaying the spread and distribution of the LPI GT data for all CRISPRi strains (ALL) and LPI GT values of CRISPRi control strains (Control). (D) Overview of the numbers of strains and genes identified as acetic acid tolerant or sensitive.

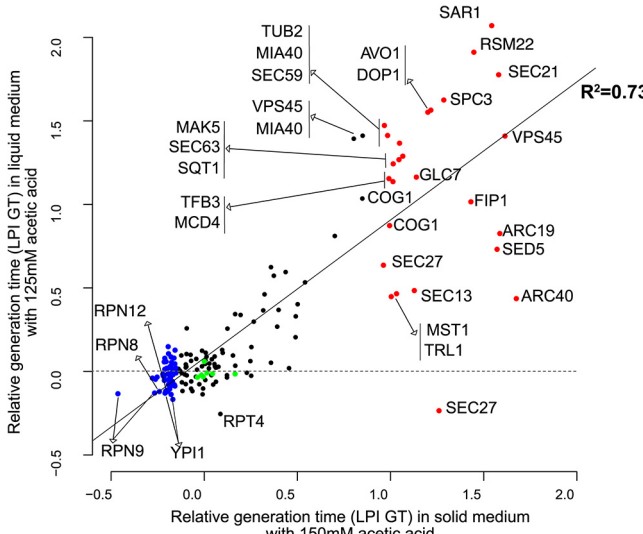

**FIG 3** Scatterplot of the relative performances of the strains in liquid medium with 125 mM acetic acid and in solid medium with 150 mM acetic acid (Scan-o-matic screening). The linear regression of the data is displayed with a black line. The means from the three LPI GT replicates for each strain are plotted, with control strains in green, acetic acid-sensitive strains in red, acetic acid-tolerant strains in blue, and the remaining strains in black. The names of the genes repressed in the tolerant or sensitive strains are indicated in the plot.

The initial screen on solid medium selected tolerant and sensitive strains based only on changes in the growth rate (generation time). In addition to the determination of the generation time, the growth analysis in liquid medium also allowed a detailed analysis of growth lag and biomass yield. A sharp reduction in the biomass yield was observed with increasing acetic acid stress (Fig. S1B). During growth in liquid medium, the generation time and yield of the strains showed strong negative correlations at both 125 mM ($r = -0.91$) and 150 mM ($r = -0.84$) acetic acid; thus, slow growth correlated with low yields during cultivation. On the other hand, neither the generation time nor the yield correlated with the lag phase, indicating that the length of the lag phase is an independent physiological feature under acetic acid stress. The lag phase of strains grown in the presence of acetic acid was much longer than that of strains grown in basal medium, whereas the changes in generation times determined were less pronounced between the two types of media. An overview of the relative performance of the strains characterized in liquid medium is demonstrated using a heat map in Fig. S2.

To investigate the relationship between the level of transcriptional repression of the target genes and the observed phenotypes, qPCR was performed for a selected set of strains with different generation times. The chosen strains had gRNAs targeting *RPN9*, *RPT4*, *GLC7*, or *YPI1* (Fig. 4 and Fig. S3). For most strains, different levels of repression of the target gene were observed using different gRNAs. For strains with gRNAs targeting *RPN9* or *GLC7*, the phenotype observed (faster growth in the case of *RPN9* and slower growth for *GLC7*) showed strong correlations with the reduction of the expression levels of the target genes ($r = 0.94$ and $r = -0.79$ for *RPN9* and *GLC7*, respectively). The expression of *GLC7* in strains with the gRNAs GLC7-TRg-2 and GLC7-NRg-4 was strongly downregulated (by ~93% and ~82%), and these two strains were also the most sensitive to acetic acid (+133% and +39% in relative generation times) (Fig. 4C). For strains with gRNAs targeting *RPT4* or *YPI1*, there was no clear correlation between the change in the expression levels and generation times (Fig. 4B and D).

**Membrane-bound organelles and vesicle-mediated secretory pathways are of particular importance under acetic acid stress.** The individual repression of 367 genes in 498 strains resulted in acetic acid sensitivity. Out of those genes, 276 are generally essential (represented by 384 strains) and 91 are respiratory growth-essential

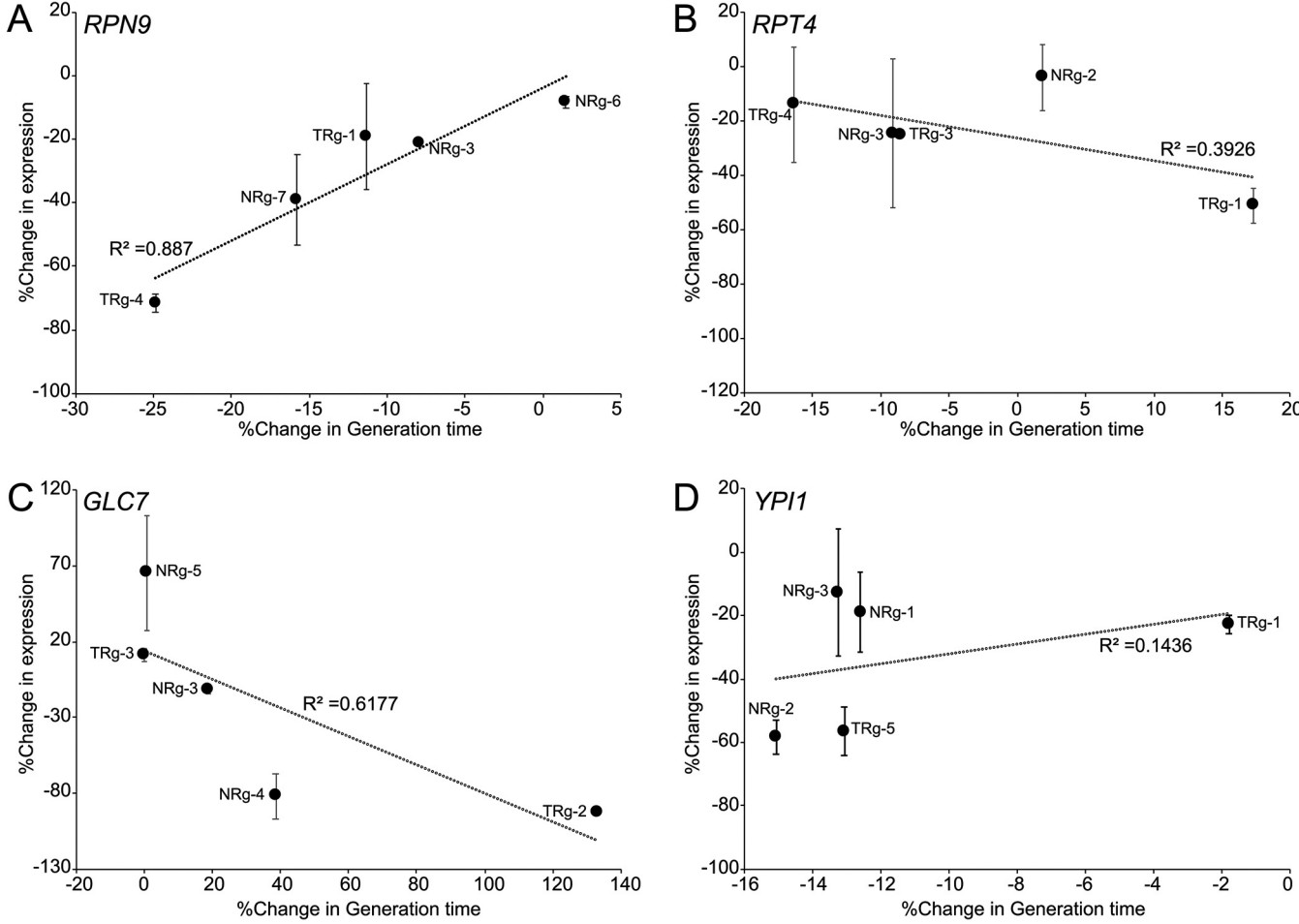

**FIG 4** Percent change in expression compared to the control strain of target genes with 125 mM acetic acid in liquid medium in relation to the percent change in relative growth of selected CRISPRi strains compared to the control strain in solid medium with 150 mM acetic acid. The gRNAs of the strains targeted *RPN9* (A), *RPT4* (B), *GLC7* (C), or *YPI1* (D). The individual points on the plot represent different gRNAs targeting the same gene. The expression of the target gene was normalized against the geometric mean values for the reference genes *ACT1* and *IPP1*. See Fig. S3 in the supplemental material for qPCR data.

(represented by 114 strains) genes (Fig. 2D) (see https://github.com/mukherjeevaskar267/ CRISPRi_Screening_AceticAcid/blob/main/BIG_DATA/Data4.xlsx).

Gene ontology (GO) enrichment analysis of genes for which repression imposed acetic acid sensitivity indicated that a fully functional bounding membrane of different organelles is of great importance to handle acetic acid stress in *S. cerevisiae* (adjusted *P* value of 0.00033) (Fig. 5). The Golgi apparatus, the endoplasmic reticulum (ER), and vesicular structures such as the endosome, the vacuole, and the organelle-associated intracellular transport pathways were found to be of particular importance (Fig. S4). Furthermore, several genes involved in vesicle-mediated transport were enriched (adjusted *P* value of 5.40e−05). Many strains with gRNAs targeting genes encoding the vacuolar membrane ATPase or GTPases required for vacuolar sorting (*VMA3*, *VMA7*, *VMA11*, *VPS1*, *VPS4*, *VPS36*, *VPS45*, or *VPS53*) were found to be sensitive to acetic acid (Table 1). Moreover, the transport of luminal and membrane protein cargoes between the ER and the Golgi segment of the secretory pathway using COPI- and COPII-coated vesicles appeared crucial for growth under acetic acid stress. Strains with gRNAs targeting genes encoding the beta′ (*SEC27*), gamma (*SEC21*), and zeta (*RET3*) subunits of the COPI vesicle coat displayed severe sensitivity to acetic acid (Table 1). Similarly, CRISPRi repression of several genes that encode components involved in the regulation of COPII vesicle coat formation (*SEC12*, *SAR1*, and *SEC23*) and COPII vesicle cargo loading (*SEC24*) and components that facilitate COPII vesicle budding (*SEC31*, *YPT1*, and *SEC13*) showed significant acetic acid sensitivity (Table 1).

 

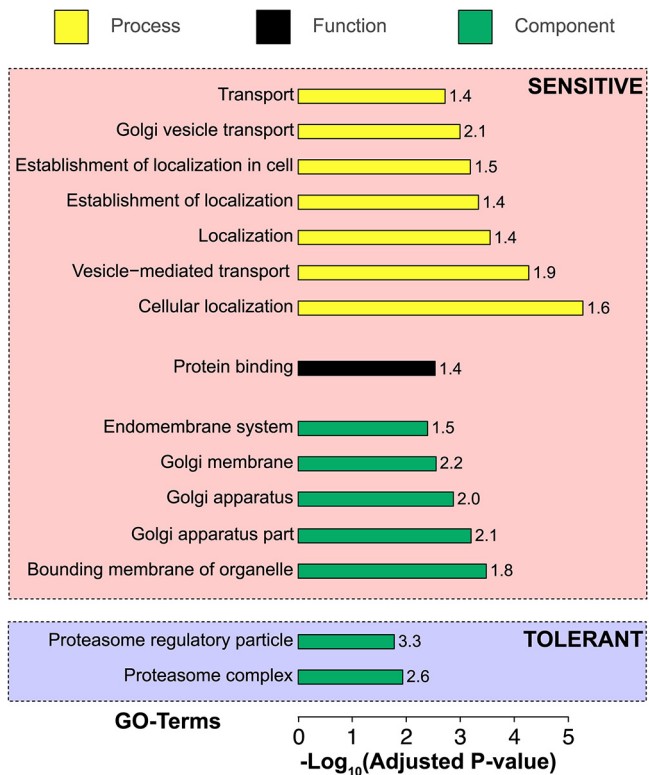

**FIG 5** Functional and gene ontology enrichment analyses of genes repressed in acetic acid-sensitive and -tolerant CRISPRi strains. GO terms connected to biological processes, genetic functions, and cell components are indicated using yellow, black, and green bars, respectively. The negative-$\log_{10}$-transformed Bonferroni-corrected $P$ values (Kruskal-Wallis test) are plotted on the $x$ axis. Enrichment factors (ratio of the observed frequency to the frequency expected by chance) for each GO term are displayed on the top of each bar.

In addition to COPI and COPII vesicle coating, our results also elucidated the importance of SNARE proteins, which mediate exocytosis and vesicle fusion with membrane-bound compartments. Our study included strains with gRNAs targeting 14 out of 24 known genes encoding SNARE proteins in *S. cerevisiae*. CRISPRi repression of 8 out of those 14 genes induced significant acetic acid sensitivity. In particular, CRISPRi repression of genes encoding v-SNARE proteins (proteins that are on the vesicle membrane) or t-SNARE proteins (proteins that are on the target membrane to which the vesicles are fused) increased the relative generation time in the presence of acetic acid (Table 1). We conclude that organelles and vesicle transport were highly enriched among sensitive strains, much in line with the findings of previous deletion collection screens, identifying that these features are important for normal growth in acetic acid (21, 22). Still, the overlap in the deletion strain collection is limited to the respiratory growth-essential genes, resulting in low overall overlap with previous findings, highlighting the need for and novelty of our screen with essential genes.

**Repression of *YPI1*, involved in the regulation of the type 1 protein phosphatase Glc7, induced acetic acid tolerance.** The accumulation of the storage carbohydrate glycogen has previously been reported to be critical for growth under acetic acid stress (23, 24). *GLC7* encodes a type 1 protein phosphatase that contributes to the dephosphorylation and, hence, the activation of glycogen synthases (25). We found that 3 out of 5 strains with gRNAs targeting *GLC7* showed significant acetic acid sensitivity, increasing the relative generation time by 16 to 120% (Fig. 4C). On the contrary, 5 strains with gRNAs targeting *YPI1*, a gene that has been reported to be involved in the regulation of Glc7, displayed significant acetic acid tolerance and reduced the relative generation time by 6 to 14% (Fig. 4D). The data obtained from solid medium were supported by the data for strains growing in liquid medium, where one strain with a gRNA

**TABLE 1** CRISPRi targeting of genes related to vesicle-, organelle-, or vesicle transport-induced acetic acid sensitivity

| Gene | Gene description[a] | No. of gRNAs | Change in generation time (%) |
|---|---|---|---|
| **Genes encoding COPI and COPII vesicle coating** | | | |
| SEC27 | Beta' subunit of COPI vesicle coat | 5 | +19 to +140 |
| SEC21 | Gamma subunit of COPI vesicle coat | 3 | +57 to complete inhibition |
| RET3 | Zeta subunit of COPI vesicle coat | 3 | +21 to +30 |
| SAR1 | Regulation of COPII vesicle coat formation | 3 | +36 to +191 |
| SEC23 | Regulation of COPII vesicle coat formation | 3 | +31 to +50 |
| SEC24 | COPII vesicle cargo loading | 2 | +27 or +72 |
| SEC13 | Facilitation of COPII vesicle budding | 2 | +25 or +119 |
| SEC12 | Regulation of COPII vesicle coat formation | 1 | +19 |
| SEC31 | Facilitation of COPII vesicle budding | 1 | +35 |
| YPT1 | Facilitation of COPII vesicle budding | 1 | +39 |
| **Genes encoding SNARE proteins** | | | |
| YKT6 | v-SNARE protein | 2 | +22 or +76 |
| BET1 | v-SNARE protein | 1 | Complete inhibition |
| BOS1 | v-SNARE protein | 1 | +18 |
| TLG1 | t-SNARE protein | 2 | +17 or +45 |
| SED5 | t-SNARE protein | 2 | +30 or +19 |
| SEC17 | Involved in SNARE complex disassembly | 1 | +29 |
| SEC22 | R-SNARE protein; assembles into SNARE complex with Bet1, Bos1, and Sed5 | 1 | +15 |
| **Vacuolar membrane ATPase complex proteins/GTPases required for vacuolar sorting** | | | |
| VPS45 | Essential for vacuolar protein sorting and also involved in positive regulation of SNARE complex assembly | 3 | +32 to +206 |
| VMA3 | Proteolipid subunit c of the V0 domain of vacuolar H$^+$-ATPase | 2 | +19 to +41 |
| VMA7 | Subunit F of the V1 peripheral membrane domain of V-ATPase | 1 | +50 |
| VMA11 | Vacuolar ATPase V0 domain subunit c | 1 | +18 |
| VPS1 | GTPase required for vacuolar sorting | 2 | +71 to complete inhibition |
| VPS4 | AAA-ATPase involved in MVB protein sorting | 1 | +21 |
| VPS36 | Involved in ubiquitin-dependent sorting of proteins into the endosome | 1 | +50 |
| VPS53 | Required for vacuolar protein sorting | 1 | +48 |

[a]MVB, multivesicular body.

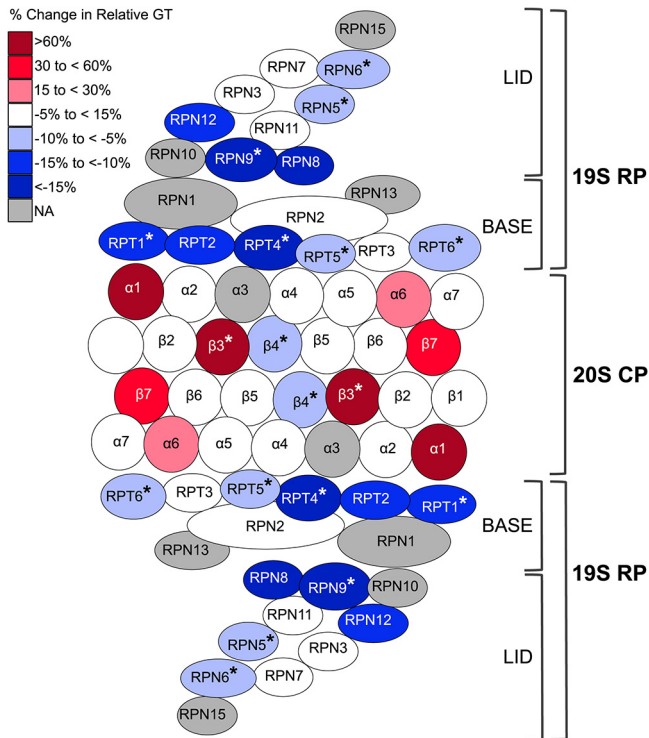

**FIG 6** CRISPRi repression of genes encoding subunits of the 26S proteasomal complex induced acetic acid tolerance (mainly genes encoding proteins of the 19S proteasomal regulatory particle lid and base subcomplexes) (displayed with blue circles) or sensitivity (genes of the 20S core particle) (displayed with red circles). The color in each subunit displays only the most dominant phenotype (i.e., significant and largest effect size) obtained by CRISPRi repression of the gene encoding that subunit. Subunits encoded by genes not included in the strain collection are displayed in gray, and subunits for which CRISPRi repression with multiple gRNAs induced the dominant phenotype are indicated with an asterisk. The schematic representation of the relative positions of the subunits in the proteasome complex is inferred from the cryo-electron microscopy (cryo-EM) structure reported previously by Luan et al. (69). NA, not applicable.

targeting *GLC7* was included. This strain showed significant acetic acid sensitivity (219% increment of the relative generation time and 42% longer lag phase) at 125 mM. In contrast, 3 out of 4 *YPI1* strains that were included in the liquid growth experiment showed significant acetic acid tolerance (11 to 13% reduction in the relative generation time and 3 to 11% reduction in the lag phase in liquid medium with 125 mM acetic acid). We tested if the effect of the repression of *YPI1* was reflected in a change in *GLC7* transcripts by qPCR analysis (data not shown). However, we scored no impact on *GLC7* mRNA levels, indicating that the potential regulation of Glc7 by Ypi1 would be at the protein level. In summary, our data give support for Ypi1 acting as a negative regulator of Glc7 at the protein level under acetic acid stress and playing an important role during growth under acetic acid conditions, possibly by affecting the accumulation of glycogen.

**The proteasome regulatory subunits have a major role in acetic acid tolerance.** Two GO terms, i.e., "proteasome complex" and "proteasome regulatory particle," were significantly enriched in the GO analysis of the 370 genes that displayed increased acetic acid tolerance when repressed by the CRISPRi system (Fig. 5). Most of the genes connected to these GO terms encode subunits of the 19S regulatory particles (RPs) of the 26S proteasome (Fig. 6 and Fig. S5). Among these were 6 genes (i.e., *RPN3*, *RPN5*, *RPN6*, *RPN8*, *RPN9*, and *RPN12*) (Table 2) encoding subunits for the RP lid assembly. CRISPRi targeting of *RPN9* was most prominent, with 5 out of 8 gRNAs inducing a significant decrease in the relative generation time, and multiple gRNAs targeting *RPN6* and *RPN5* also induced acetic acid tolerance (Table 2). Overall, the different gRNAs for

**TABLE 2** CRISPRi targeting of genes encoding proteins of the 19S proteasomal regulatory particle lid and base subcomplexes induced acetic acid tolerance

| Gene | Gene description | No. of gRNAs | Change in generation time (%) |
|---|---|---|---|
| Proteasome 19S regulatory particle lid complex | | | |
| RPN9 | Non-ATPase regulatory subunit of the 26S proteasome lid | 5 | −5 to −27 |
| RPN6 | Non-ATPase regulatory subunit of the 26S proteasome lid; required for the assembly and activity | 3 | −4 to −8 |
| RPN5 | Non-ATPase regulatory subunit of the 26S proteasome lid | 2 | −2 or −6 |
| RPN3 | Non-ATPase regulatory subunit of the 26S proteasome lid | 1 | −3 |
| RPN8 | Non-ATPase regulatory subunit of the 26S proteasome lid | 1 | −15 |
| RPN12 | Non-ATPase regulatory subunit of the 26S proteasome lid | 1 | −14 |
| Proteasome 19S regulatory particle base complex | | | |
| RPT1 | ATPase of the 19S regulatory particle | 3 | −5 to −12 |
| RPT4 | ATPase of the 19S regulatory particle | 2 | −11 to −16 |
| RPT5 | ATPase of the 19S regulatory particle | 2 | −3 to −4 |
| RPT6 | ATPase of the 19S regulatory particle | 2 | −6 to −7 |
| RPT2 | ATPase of the 19S regulatory particle | 1 | −11 |
| Proteasome 20S core particle | | | |
| SCL1 | Alpha 1 subunit of the 20S proteasome | 1 | +74 |
| PRE5 | Alpha 6 subunit of the 20S proteasome | 1 | +15 |
| PRE4 | Beta 7 subunit of the 20S proteasome | 1 | +31 |
| PUP3 | Beta 3 subunit of the 20S proteasome | 2 | +30 to +63 |

these different RP lid assembly genes reduced the relative generation time in the range of 2 to 27% (Table 2).

The performances of 10 strains with gRNAs targeting subunits of the 19S regulatory particle lid complex were also characterized in liquid medium. Both strains with gRNAs inducing tolerance and strains with gRNAs failing to give a measurable phenotype on solid medium were included. Most of the strains (4 out of 6) identified as tolerant on solid medium (with gRNAs targeting *RPN9* or *RPN12*) also showed significant acetic acid tolerance in liquid medium, with an 8 to 12% reduction in the relative generation time and a 4 to 8% reduced lag phase at 125 mM acetic acid (Fig. 7A).

In addition to acetic acid tolerance achieved by targeting the lid of the 19S regulatory particle, several CRISPRi strains targeting genes encoding subunits of the 19S RP base assembly showed significant acetic acid tolerance (Fig. 6 and Table 2). A reduction of 3 to 12% of the relative generation time was observed for strains with gRNAs targeting the RP base assembly subunit *RPT1*, *RPT2*, *RPT4*, *RPT5*, or *RPT6*. The fitness benefit of targeting *RPT4* was confirmed in liquid medium, where the strain RPT4-NRg2 (Fig. 7A) had a 22% reduced relative generation time at 125 mM acetic acid.

In contrast to the increased tolerance seen when targeting the 19S regulatory particle, CRISPRi targeting of genes encoding the 20S proteasome predominantly led to acetic acid sensitivity (Fig. 6). The relative generation times were increased by 15 to 74% in strains with gRNAs targeting *SCL1*, *PRE5*, *PRE4*, or *PUP3* (Table 2). This trend was confirmed in liquid medium, where 6 out of 11 strains with gRNAs targeting genes encoding 20S proteasomal subunits showed significant acetic acid sensitivity (Fig. 7A). Thus, our data indicate that the proteasome and its different subparts play critical and differential roles in regulating growth in medium with acetic acid.

## DISCUSSION

**Bioengineering of essential genes in yeast using CRISPRi technology.** A number of large-scale, systematic, gene-by-phenotype analyses of essential genes have previously been performed by phenotyping either heterozygous deletion mutants or strains carrying temperature-sensitive alleles (26–29). Nonetheless, the use of heterozygous deletion mutants is limited by haplosufficiency, as one copy of a gene is often adequate for the normal function of diploids (30). Moreover, temperature-dependent side effects may influence the results when studying thermosensitive alleles (28, 31). For a subset of essential genes, systematic studies employing tetracycline-regulatable repressor (TetR) system-based promoter engineering in *S. cerevisiae* have been reported (3, 32, 33), but this study represents the first example where >99% of the essential genes were individually screened.

In previous studies where CRISPRi technology was applied for massive genotype-phenotype mapping in *S. cerevisiae* (9, 11–13), the strains were pooled and screened for competitive growth. Although competitive growth assays have the advantage of throughput, they come with a major weakness: the nutrient-specific advantage for cells/strains with a shorter lag phase is amplified. Single-cell analysis has shown massive heterogeneity in lag phases within clonal populations of *S. cerevisiae* (34), which may introduce noise in the outcome of competitive growth assays. Moreover, the characterization of a population enriched after a specific time provides merely an endpoint observation. In the previously described competitive growth assays of whole-genome CRISPRi libraries (11, 13), the genes identified to give beneficial phenotypes when repressed have not been essential. This is likely due to the phenotypes of strains with altered expression of essential genes not being as pronounced as the phenotypes of the strains becoming enriched or due to the alteration in expression being detrimental. Often, the genetic or environmental effects on cellular fitness are relatively small (35, 36), and thus, highly accurate measurement methodologies are required to capture subtle differences in growth phenotypes. Therefore, we used the Scan-o-matic phenomics platform (20) to individually grow each of the >9,000 strains of a CRISPRi strain library. The generation time of each strain was generated from high-resolution growth curves without the influence of or competition from other strains.

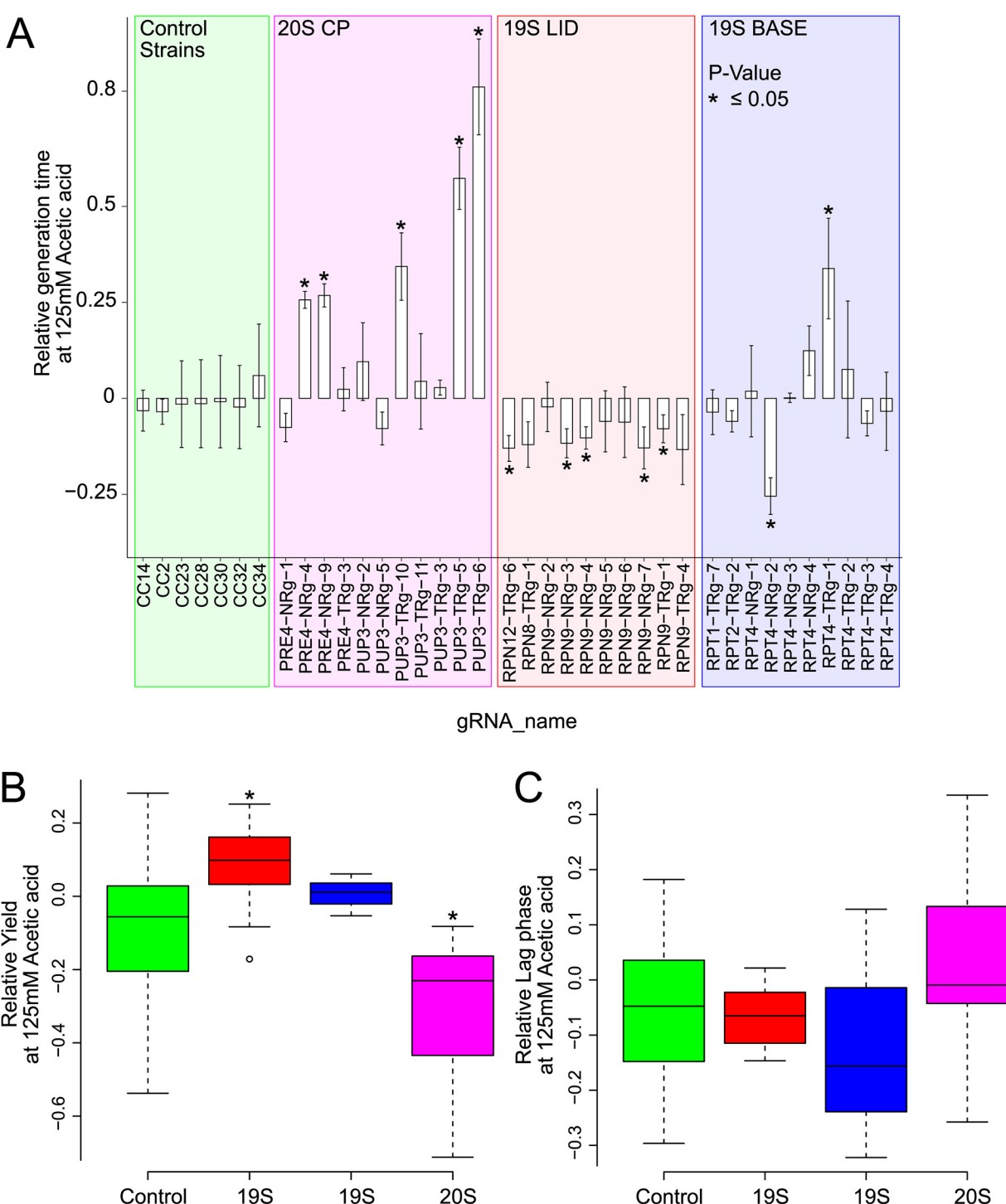

**FIG 7** (A) Relative growth in liquid medium of CRISPRi strains with gRNAs targeting genes encoding proteasomal subunits (20S CP [core particle], 19S lid, or 19S base) and control strains. The relative generation times of all strains (A) and biomass yields (B) and lag phases (C) of the acetic acid-tolerant strains are shown.

During growth under basal conditions, we found that most of the CRISPRi strains grew with a generation time similar to or just slightly lower than the generation time of the control strains. In medium with acetic acid, there was great variability between the strains, with some growing faster and, as expected, many growing much slower. Only about 1% of the strains of the library did not grow under basal conditions. This in

line with what Smith et al. (12) observed when growing pooled strains in yeast extract-peptone-dextrose (YPD) medium: after 10 doublings, the DNA barcodes associated with 170 strains dropped below the background. In basal medium, most strains grew rapidly, and it may be that for some strains, the CRISPRi repression that was induced only at the start of the screen did not reach the critical threshold needed to induce growth defects, as was recently shown in an *Escherichia coli* CRISPRi screen, where it was proposed that cells have developed robustness against somewhat changing protein levels (37).

Our qPCR profiling of selected genes of strains during mid-exponential growth showed that under both basal conditions and acetic acid stress, different levels of repression were achieved by targeting the same gene with different gRNAs (Fig. 4). For the tested genes, we observed that the repression of expression was more pronounced in basal medium than in medium supplemented with acetic acid (see Fig. S3 in the supplemental material), indicating that repression by the CRISPRi system may be influenced by environmental conditions. High concentrations of acetic acid are known to cause an increased lag phase (38). We observed that several of the strains scored for a change in the growth rate also displayed defects or improvements in the length of the lag phase, while some did not (Fig. S2).

**CRISPRi targeting of genes related to vesicle, organelle, or vesicle transport causes acetic acid sensitivity.** Previous large-scale screens of strains have identified many genes with widely diverse functions, the deletions of which increased the susceptibility of yeast to acetic acid (22, 39). In line with our findings, Sousa et al. (22) reported that deleting genes involved in vesicular traffic from the Golgi apparatus to the endosome and the vacuole increased sensitivity to acetic acid. In addition, endocytic inhibition has been observed in response to acetic acid and other environmental stressors (40). Many of the acetic acid-sensitive strains in our study had gRNAs targeting genes encoding different proteins involved in the formation and activity of COPI and COPII vesicles or SNARE proteins (Table 1). The COPI and COPII vesicles transport proteins between the ER and the Golgi apparatus (reviewed in reference 41), whereas SNARE proteins mediate exocytosis and vesicle fusion with different membrane-bound compartments (reviewed in reference 42). It has been reported that acetic acid causes ER stress and induces the unfolded protein response, as misfolded proteins accumulate in the ER (43). A previous study screening the deletion strain collection reported ER, Golgi, and vacuolar transport processes as being important for resistance to a vast collection of small molecules or environmental stress conditions, including acetic acid treatment (44).

Deletions of genes encoding the vacuolar membrane ATPase complex (*VMA2-8, -13, -16, -21,* and *-22*) have been shown to decrease tolerance to acetic acid (22, 39), presumably as cells struggle to maintain a neutral cytosolic pH (45). Similarly, single-gene deletions of vacuolar protein sorting (VPS) genes (encoding GTPases required for vacuolar sorting) have been shown to result in drastically enhanced sensitivity to acetic acid and a drop in the intracellular pH (46). In line with these studies, we found strains with gRNAs targeting several vacuolar ATPase-related genes (encoding vacuolar membrane ATPase [VMA] and VPS complexes) (Table 1) to be among the sensitive strains, highlighting the importance of the vacuole in the response to acetic acid stress.

**Regulation of genes involved in glycogen accumulation influences acetic acid tolerance.** Glycogen serves as a fuel reserve for cells and accumulates when growth conditions deteriorate as a means of adapting to stress such as nutrient, carbon, or energy limitation (47) or acetic acid treatment (23, 24). Glycogen is produced from glucose-6-phosphate via glycogen synthases that are activated by dephosphorylation by, e.g., the Glc7 phosphatase (25).

Hueso et al. (48) demonstrated that the overexpression of a functional, 3′-truncated version of the *GLC7* gene improved acetic acid tolerance. In our study, 3 strains with gRNAs targeting *GLC7* showed strong acetic acid sensitivity (Fig. 4C). Ypi1 was initially reported to be an inhibitor of Glc7 (49), while it was later shown to positively regulate Glc7 activity in the nucleus (50). The overexpression of *YPI1* has been shown to reduce

glycogen levels (49). Our study showed that the downregulation of *YPI1*, encoding a regulatory subunit of the type 1 protein phosphatase Glc7, conferred acetic acid tolerance. Five strains with gRNAs targeting *YPI1* displayed significant decreases in generation times when subjected to acetic acid, and the downregulation of *YPI1* in these CRISPRi strains was confirmed by qPCR (Fig. 4D).

In light of the fact that both Ypi1 and Glc7 have many different roles in maintaining cell homeostasis beyond glycogen synthesis, we propose that CRISPRi-mediated repression of *YPI1* may be favorable for the cells under acetic acid stress, likely due to increased glycogen levels in the cells. Similarly, we suggest that CRISPRi strains where *GLC7* is repressed may have decreased intracellular glycogen contents, thus rendering them more sensitive to acetic acid. Still, it may be that other regulatory roles of Ypi1 and Glc7 are behind the acetic acid resistance/sensitivity identified for some of the CRISPRi strains, and determination of this needs further study.

**Adapting proteasomal degradation of oxidized proteins may save ATP and increase acetic acid tolerance.** While the best-known function of the proteasome is ATP-dependent protein degradation through the 26S ubiquitin-proteasome system, the unbound, ATP-independent 20S proteasome is the main protease responsible for degrading oxidized proteins (reviewed in reference 51). The 26S proteasomal complex consists of one 20S core particle (CP) and two 19S regulatory particles that are further divided into lid and base assemblies. In our study, many of the strains with increased acetic acid tolerance had gRNAs targeting genes encoding subunits of the 19S regulatory particle of the proteasome (Fig. 6 and Fig. 7A). Notably, none of the acetic acid-tolerant strains with gRNAs targeting the 19S regulatory particle of the proteosome showed a severe growth defect in basal medium (https://github.com/mukherjeevaskar267/CRISPRi _Screening_AceticAcid/blob/main/BIG_DATA/Data2.xlsx), indicating that they are potentially good target strains in industrial applications.

Many studies have reported the accumulation of reactive oxygen species (ROS) under acetic acid stress, and reactive oxygen species are well known to cause protein oxidation and even induce programmed cell death in cells upon acetic acid stress (reviewed in reference 52). Yeast cells under oxidative stress respond to the accumulation of ROS with a decrease in the cellular ATP concentration (53). Acetic acid that enters the cell dissociates to protons and acetate ions at a nearly neutral cytosolic pH, and the charged acetate ions are unable to diffuse through the plasma membrane and thus accumulate intracellularly (reviewed in reference 45). Therefore, acetic acid stress, in particular pumping out excess protons from the cytosol to the extracellular space by $H^+$-ATPase pumps in the plasma membrane and from the cytosol to the vacuole by the vacuolar $H^+$-ATPases, causes a reduction in ATP (45). Moreover, the accumulation of ROS has been reported to induce a metabolic shift from glycolysis to the pentose phosphate pathway in order to increase the production of NADPH, an essential cofactor to run the antioxidant systems, which leads to a reduction in ATP generation (54). Consequently, ATP conservation by reducing the activity of ATP-dependent processes could offer yeast a fitness benefit against acetic acid stress.

The 20S core particle on its own performs ubiquitin- and ATP-independent degradation of proteins. Under acetic acid stress, ROS accumulation triggers protein oxidation that leads to protein unfolding (55). The inner proteolytic chamber of the 20S core particle is accessible to only unfolded proteins, and moderately oxidized proteins are ideal substrates for the 20S proteasome (56–58). We hypothesize that the repression of subunits of the 19S regulatory particle increases the abundance of free 20S core particles, which offers the cell an alternative to ATP-expensive 26S proteasome-mediated protein degradation. In line with this, it has been reported that even mild oxidative stress reversibly inactivates both the ubiquitin-activating/conjugating system and 26S proteasome activity but does not impact the functionality of the 20S core particle (59, 60). Therefore, an increased abundance of the 20S core particle alone in strains where the CRISPRi system targets genes encoding subunits of the 19S regulatory particle could allow more efficient ATP-independent degradation of oxidized proteins, thus conferring a fitness benefit to yeast during acetic acid stress (Fig. 8).

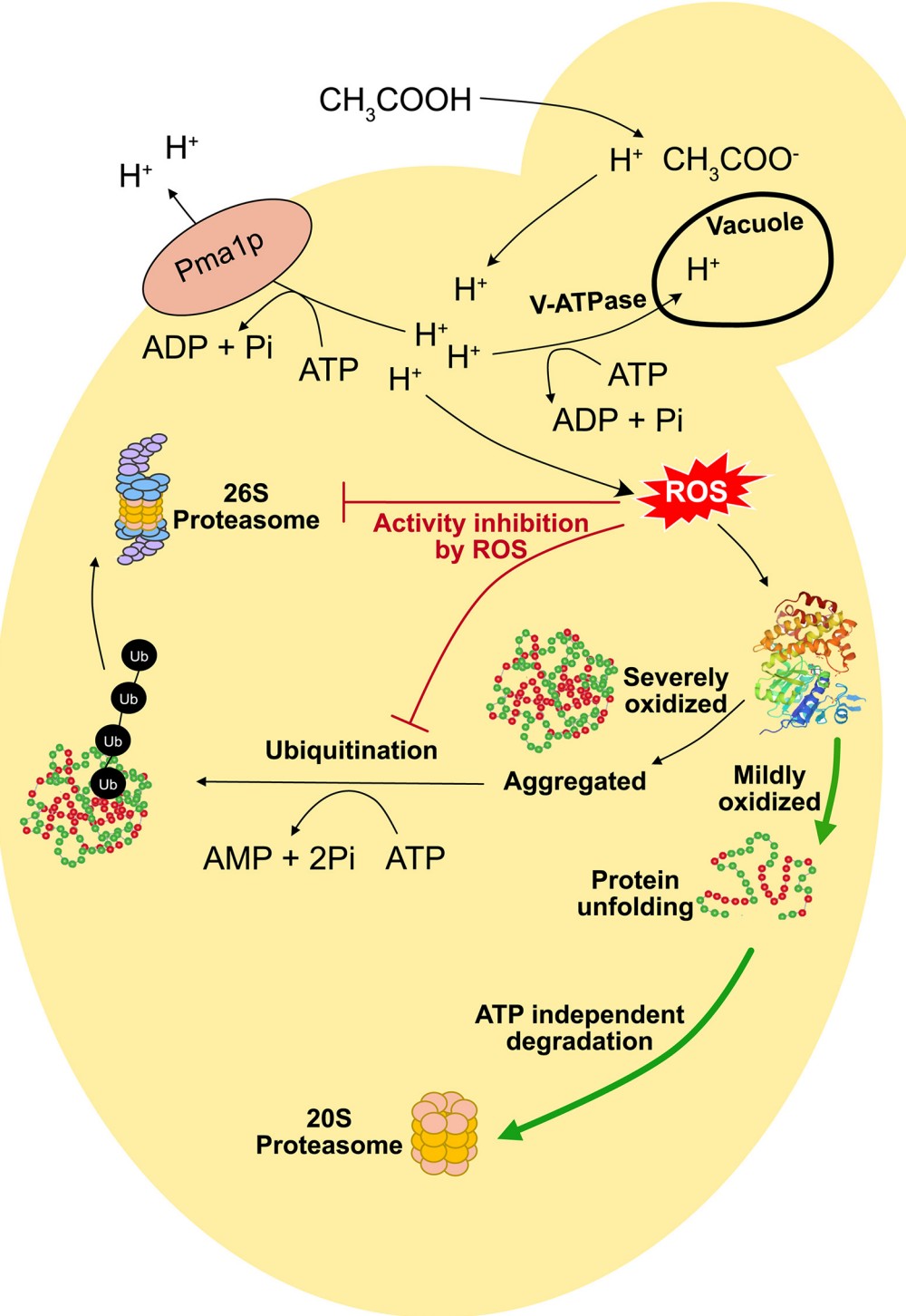

**FIG 8** Overview of the response of the cells to acetic acid stress based on CRISPRi targeting of essential genes. The cells are starved of ATP due to ATP-expensive processes such as the elevated action of H⁺-ATPase and V-ATPase pumps. Therefore, we hypothesize that the downregulation of subunits of the 19S RP increases the abundance of the 20S CP, which offers the cell an alternative to ATP-expensive 26S proteasome-mediated protein degradation. This in turn gives yeast a fitness benefit under oxidative stress induced by acetic acid. ROS, reactive oxygen species; Ub, ubiquitin.

A total of five CRISPRi strains with gRNAs targeting *RPN9* (encoding a subunit of the 19S regulatory lid assembly) had significantly decreased generation times in medium supplemented with acetic acid (Table 2). This gives confidence that the downregulation of *RPN9* provides a means to improve acetic acid tolerance. Previously, an *rpn9*

mutant with defective assembly of the 26S proteasome and reduced 26S proteasome activity was shown to be more resistant to hydrogen peroxide, which is a common stressor used to enforce oxidative stress (61). Moreover, this *rpn9* mutant was able to degrade carbonylated (oxidized) proteins more efficiently than the wild-type strain, and it displayed increased 20S-dependent proteasome activity (61). In our study, we observed that the yields of strains with gRNAs targeting the 19S lid or base of the proteasome were increased for strains growing in acetic acid, whereas the yields of strains with gRNAs targeting the 20S CP of the proteasome were decreased (Fig. 7B). It seems plausible that the repression of genes encoding subunits of the 19S lid leads to decreased ATP-expensive 26S activity and that this ATP savings contributed to a concomitant increment in biomass.

Our qPCR results showed that the level of repression of *RPN9* or *RPT4* (encoding a subunit of the 19S regulatory base assembly) was greatly dependent on the gRNA of the strains (Fig. 4A and B). For *RPN9*, there was a strong correlation between expression levels and acetic acid tolerance, indicating that fine-tuning 20S and 26S proteasomal regulation could be an efficient strategy to bioengineer acetic acid-tolerant industrial yeast strains (Fig. 4A). In line with this, a recent study showed that the downregulation of *RPT5* (encoding a subunit of the 19S base assembly) induced tolerance to oxidative stress (62). In our study, the downregulation of *RPT4* was shown to improve acetic acid tolerance for 3 out of 5 strains with gRNAs targeting this gene (Fig. 4B and Fig. S3B). Nonetheless, the generation time of RPT4-TRg-1 with clear repression of *RPT4* was increased. We argue that a too-strong repression of an essential gene is likely to be detrimental, highlighting the need for fine-tuned expression when engineering tolerance. While off-target effects of gRNAs as well as gRNAs failing to give a phenotype are known challenges of the CRISPRi technology, screening several strains with different gRNAs and identifying multiple strains with similar phenotypes give confidence in a phenotype being a result of the gene repression itself (12). In our study, a total of 28 strains with gRNAs targeting proteasomal genes were identified as tolerant or sensitive (Table 2), which gives great confidence for us to elaborate on the role of the proteasome during acetic acid stress.

In conclusion, our study identified many essential and respiratory growth-essential genes that regulate tolerance to acetic acid. CRISPRi-mediated repression of genes involved in vesicle formation or organelle transport processes led to severe growth inhibition during acetic acid stress, emphasizing the importance of these intracellular membrane structures for maintaining cell vitality. The data also suggest that increased activity of ATP-independent protein degradation by the 20S core is an efficient way of counteracting acetic acid stress. This mechanism may ensure ATP savings, allowing proton extrusion and an increased biomass yield. Fine-tuned expression of proteasomal genes could be a strategy for increasing the stress tolerance of yeast, leading to improved strains for the production of biobased chemicals.

## MATERIALS AND METHODS

**Yeast strain library.** The CRISPRi strain library (12) used in this study contains 9,078 strains, each of which has an integrated dCas9-Mxi1 repressor (14). The strains also contain a tetracycline-regulatable repressor (TetR), where TetR controls a modified polymerase III (Pol III) promoter (TetO-PRPR1) that drives the expression of unique gRNAs (Fig. 1). Thus, the gRNAs are expressed in the presence of the inducing agent anhydrotetracycline (ATc). Each strain in this library expresses a unique gRNA that, in combination with dCas9-Mxi1, targets 1,108 of the 1,117 (99.2%) essential genes (30) and 505 of the 514 (98.2%) respiratory growth-essential genes (63, 64) in *S. cerevisiae* (see Fig. S6A and B in the supplemental material). For most of the genes (1,474 out of 1,617), there are at least 3 and up to 17 strains (mean, ~5) with different gRNAs targeting the same gene in the library (Fig. S6C). A total of 93% of the unique gRNAs were designed within 200 bp upstream of the transcription start site of the respective target gene (Fig. S6D). Depending on the targeting location of the gRNA in the promoter, genetic repression ranging from very strong to weak can be achieved (8). This produces strains that under ATc induction have different levels of repression of the same gene relative to the native expression level. Moreover, 20 strains in the CRISPRi library have gRNAs that are nonhomologous to the *S. cerevisiae* genome and function as control strains (Fig. S6B). The CRISPRi strains were stored in a yeast extract-peptone (YP)-glycerol stock solution (17% [vol/vol] glycerol, 10 g/liter yeast extract, 20 g/liter Bacto peptone). The whole

collection was kept in 24 microtiter plates (MTPs) (384-well format). Unless otherwise mentioned, all chemicals were purchased from Merck.

**ATc titration in YNB medium.** Synthetic defined medium was used to identify acetic acid-specific effects, excluding compounds present in rich medium that might confound the interpretation of our data. To obtain appropriate gene suppression in our setup, we adjusted the concentration of ATc in relation to what had been proposed previously for rich-medium liquid cultures (12).

The concentration of ATc sufficient to induce a high level of gRNA expression in the CRISPRi strains growing on yeast nitrogen base (YNB) agar medium was determined by a qualitative spot test with selected strains (Fig. S7A). These strains were selected based on the competitive growth assay of the CRISPRi library in liquid YPD medium with and without 250 ng/ml of ATc by Smith et al. (12). This study showed that the growth of strains with gRNAs targeting the essential genes *ACT1* (ACT1-NRg-5, TTAAACAAGAGAGATTGGGA; ACT1-NRg-8, ATTTCAAAAAGGAGAGAGAG), *VPS1* (VPS1-TRg-1, GCCGGGT CACCCAAAGACTT), and *SEC21* (SEC21-NRg-5, GTCGTAGTGAATGACACAAG) was nearly or completely inhibited, as these essential genes, targeted by the gRNAs of the strains, were strongly repressed. These strains as well as two control strains, i.e., Ctrl_CC11 (CC11, CCCAGTAGCTGTCGGTAGCG) and Ctrl_CC23 (CC23, AGGGGTGCTAGAGGTTTGCG), were grown on synthetic defined YNB agar medium (1.7 g/liter yeast nitrogen base without amino acids and ammonium sulfate [BD Difco], 5 g/liter ammonium sulfate, 0.79 g/liter complete supplement mixture with all amino acids and nucleotides [Formedium], 20 g/liter glucose, 20 g/liter agar, and succinate buffer, i.e., succinic acid at 10 g/liter and sodium hydroxide at 6 g/liter) in the presence of 0, 2.5, 5, 7.5, 10, 12.5, or 25 $\mu$g ATc/ml. A stock solution (25 mg/ml in dimethyl sulfoxide [DMSO]) was used to achieve the different ATc concentrations. The final concentration of DMSO in the medium was adjusted to 0.1% (vol/vol). The precultures for the spot assay were grown in liquid YNB medium for 48 h, after which 3-$\mu$l drops from serial dilutions ($10^{-1}$, $10^{-2}$, $10^{-3}$, and $10^{-4}$) of a cell suspension at an optical density at 600 nm ($OD_{600}$) of 1 were spotted on solid YNB medium with different concentrations of ATc and incubated at 30°C for 48 h. We found that 2.5 $\mu$g/ml of the gRNA inducer ATc was sufficient to elicit growth defects on solid medium for strains with gRNAs targeting the essential gene *ACT1*, *VPS1*, or *SEC21*. The growth of these strains was incrementally inhibited up to nearly complete inhibition at 7.5 $\mu$g/ml ATc (Fig. S7A). In contrast, the growth of the control strains (strains expressing gRNAs with no genomic target) remained unimpeded even at 25 $\mu$g/ml ATc (Fig. S7A), and we therefore used 7.5 $\mu$g/ml ATc in our screen of the CRISPRi library.

A liquid ATc titration assay was done in 200 $\mu$l liquid YNB medium with 0, 0.25, 1, 2, 3, 5, 7.5, 10, 15, or 25 $\mu$g ATc/ml in a Bioscreen C microbiology reader device (Fig. S7B). The strains were precultured in YNB medium for 48 h. A separate preculture was used to inoculate each replicate at a starting $OD_{600}$ of approximately 0.1. In order to avoid an uneven oxygen distribution, the plastic cover of the Bioscreen plate was replaced with a sterile sealing membrane permeable to oxygen, carbon dioxide, and water vapor (Breathe-Easy; Sigma-Aldrich). Strains were grown with continuous shaking for 75 h, during which automated spectrophotometric readings were taken every 20 min. The raw data were calibrated to actual $OD_{600}$ values and smoothed before the growth lag, generation time, and growth yield were estimated using PRECOG software (65). All growth experiments were performed at 30°C.

**Medium preparation for high-throughput phenomics.** Solid YPD medium (10 g/liter yeast extract, 20 g/liter Bacto peptone, 20 g/liter glucose, 20 g/liter agar) was used to regrow the CRISPRi collection from −80°C storage and also to grow the precultures. The growth phenotypes of all the CRISPRi strains in the library were evaluated under basal conditions, i.e., solid YNB medium and solid YNB medium supplemented with 150 mM acetic acid. ATc (7.5 $\mu$g/ml, as determined by the qualitative spot assay) was added to both media to induce gRNA expression. The acetic acid concentration used was determined by growing a subset of the CRISPRi strains (739 strains), which were pinned to the actual experimental format of 1,536 colonies per plate, on solid medium with different acetic acid concentrations (50, 75, 100, and 150 mM). The largest phenotypic difference in growth between the strains was observed at 150 mM acetic acid (Fig. S8A), and this concentration was selected to be used in our screen. The final concentration of DMSO in the growth medium was 0.03% (vol/vol), and the pH was adjusted to 4.5.

**High-throughput phenomics using Scan-o-matic.** The high-throughput growth experiments were performed using the Scan-o-matic (20) phenomics facility at the University of Gothenburg, Sweden. The procedure is described here in short. A robotic Singer high-density array rotor was used for all replica pinning. First, the −80°C frozen stock of the CRISPRi library in 24 microtiter plates was pinned in a 384-well array format on solid YPD medium and then incubated at 30°C for 72 h in scanners imaging the plates. For each of the 24 plates, one preculture plate was prepared in a 1,536-well array format. For this purpose, 384 strains were pinned three times so that each had 3 adjacent replicates. In this way 384 × 3, i.e., 1,152, positions in a 1,536-well array format were filled. All fourth positions, i.e., the rest of the 384 positions, were filled with a spatial control strain to normalize any spatial growth bias (Fig. S8B). The Scan-o-matic system uses a dedicated algorithm that can normalize any spatial growth bias in the extracted phenotypes of the other strains using the growth data of this spatial control strain (20). Here, the control strain Ctrl_CC23 was used as the spatial control strain. The preculture plates were incubated at 30°C for 48 h before being used for replica pinning on the experimental plates, which were placed in the scanners in a predefined orientation and incubated at 30°C. The plates were imaged automatically every 20 min for 96 h. Subsequently, image analysis by Scan-o-matic was performed, and a growth curve was generated for each colony. Finally, absolute and spatially normalized generation times were extracted for all replicates of each strain. The whole experimental process was repeated twice to generate 6 experimental replicate measurements for each strain in both the medium with 150 mM acetic acid and the basal medium lacking acetic acid.

mSystems®

**Data analysis.** R version 4.0.2 was used to perform all mathematical and statistical analyses. The analytical steps employed to identify essential or respiratory growth-related genes that lead to acetic acid sensitivity or tolerance when repressed are described below. Here, we also explain the terminologies used.

**(i) Normalized generation time and batch correction.** The normalized generation time obtained after the spatial bias correction gives the population doubling time of a strain colony relative to the spatial control strain on a $\log_2$ scale (20). This is referred to as the log strain coefficient for the generation time (LSC GT). In previous studies, it was found that only a few of the gRNAs targeting a specific gene can induce strong repression that results in a strong phenotypic effect (8, 12), and therefore, most of the strains will display a phenotype similar to that of a control strain. Since we used the control strain Ctrl_CC23 as the spatial control strain, it was expected that the median LSC GT of all strains in an experimental plate would be close to zero. However, some variability in the data set was still present due to unavoidable microenvironmental factors between plates, and this caused a slight deviation of the median value of the LSC GT for some experimental plates. To correct for this batch effect, plate-wise correction was conducted by subtracting the median of the LSC GT values of all the individual colonies on a plate from the individual LSC GT values of the colonies growing on that plate; i.e., if strain $X$ is growing on plate $Z$, the corrected LSC GT value for strain $X$ is as follows: LSC GT_corrected$_{\text{strain } X}$ = (LSC GT$_{\text{strain}}$) − median(LSC GT_plate $Z$).

**(ii) Relative generation time in the presence of acetic acid.** The growth of each CRISPRi strain was evaluated under two different conditions, i.e., medium with 150 mM acetic acid (AA$_{150\text{ mM}}$) and basal medium lacking acetic acid (basal.condition). The relative performance of a strain in the presence of acetic acid compared to the basal conditions was determined by subtracting LSC GT_basal.condition from LSC GT_AA$_{150\text{ mM}}$. This relative estimation, which gives the acetic acid-specific effect on the generation time (GT) of a strain, is defined as the log phenotypic index (LPI GT) (66); i.e., for strain $X$, the LPI GT was calculated as follows: LPI GT = LSC GT_AA$_{150\text{ mM\_strain } X}$) − (LSC GT_basal.condition$_{\text{strain } X}$).

**(iii) Statistical tests and $P$ value adjustment.** Since it was expected that most strains would show only minor changes in the generation time, here, it is hypothesized that a phenotypic difference between a specific CRISPRi strain and the mean phenotypic performance of all the CRISPRi strains that falls within the interquartile range (IQR) of the complete data set (i.e., having an LPI GT value of between −0.024 and 0.075) would be zero and any difference within the IQR to be just by chance. Therefore, formally, our null hypothesis ($H_0$) was $\mu\_$strain $X$(all_replicates_LPI GT) − $\mu$(IQR_LPI GT) = 0; i.e., the difference between the mean LPI GT of all replicates of strain $X$ and the mean of the LPI GT data set within the IQR equals zero. The $P$ value for each strain in the library was estimated using Welch's two-sample two-sided $t$ test, which is an adaptation of Student's $t$ test and produces fewer false positives (67). Moreover, this method remains robust for skewed distributions and large sample sizes. In this study, the mean LPI GT of 3,392 strains displayed a significant ($P$ value of ≤0.1) deviation from $\mu$(IQR_LPI GT) when subjected to Welch's two-sample two-sided $t$ test (Fig. S9A). The $P$ values were corrected by the Benjamini-Hochberg method, also known as the false discovery rate (FDR) method (68). An adjusted $P$ value threshold of ≤0.1 was set to select acetic acid-tolerant or -sensitive strains. The application of the FDR method (68) left 1,258 strains below the adjusted $P$ value threshold of 0.1 (Fig. S9B). None of the control strains had an adjusted $P$ value below 0.1 (Fig. S9D).

An LPI GT threshold was applied for the selection of tolerant or sensitive strains. If a CRISPRi strain had a mean LPI GT that was greater than the maximum of the mean LPI GT of the control strains, then the strain was considered acetic acid sensitive. Similarly, if a CRISPRi strain had a mean LPI GT that was less than the minimum of the mean LPI GT of the control strains, then the strain was considered acetic acid tolerant. In this study, we observed that the range of mean LPI GT values for the control strain was between −0.037 and 0.166.

Therefore, acetic acid-sensitive strain = $\mu_{\text{strain}}$(LPI GT) > 0.166 and adjusted $P$ ≤ 0.1, and acetic acid-tolerant strain = $\mu_{\text{strain}}$(LPI GT) < −0.037 and adjusted $P$ ≤ 0.1.

Some CRISPRi strains that grew well under basal conditions but very poorly or not at all on the acetic acid experimental plates were identified. These strains were not subjected to any statistical analysis but were still added to the final list of acetic acid-sensitive CRISPRi strains.

**Gene ontology analysis.** Gene ontology (GO) term (process, function, and component) enrichment analysis of the gene lists of acetic acid-tolerant and -sensitive strains was performed against a background set of genes (all 1,617 genes targeted in this CRISPRi library) using the GO term finder in the *Saccharomyces* genome database (version 0.86) (https://www.yeastgenome.org/goTermFinder), and all GO term hits with $P$ values of <0.1 were identified.

**Growth of selected strains in liquid medium.** In order to validate the acetic acid sensitivity or tolerance observed for the CRISPRi strains in the Scan-o-matic screening, selected strains were grown in liquid YNB medium using the Bioscreen platform. The 48 most acetic acid-sensitive and the 50 most tolerant CRISPRi strains from the Scan-o-matic analysis were selected for validation. Moreover, all CRISPRi strains with gRNAs targeting any of the 12 genes *RPT4*, *RPN9*, *PRE4*, *MRPL10*, *MRPL4*, *SEC27*, *MIA40*, *VPS45*, *PUP3*, *VMA3*, *SEC62*, and *COG1* were included, making a total of 176 strains that were grown together with 7 control strains in liquid medium (raw data are available at https://github.com/mukherjeevaskar267/CRISPRi _Screening_AceticAcid/blob/main/BIG_DATA/Data5.xlsx).

Briefly, the strains were pinned from the frozen stock into liquid YNB medium and grown at 30°C for 40 h at 220 rpm. This plate was used as the preculture, and separate precultures were prepared for each independent culture. The strains were grown in liquid YNB medium (basal conditions) and liquid YNB medium supplemented with 125 or 150 mM acetic acid. For each strain, 3 independent replicates were included for each growth condition. Two micrograms per milliliter of ATc was added

to the medium to induce gRNA expression. The final concentration of DMSO in the growth medium was 0.008% (vol/vol), and the pH was adjusted to 4.5. The experimental method and subsequent phenotype extraction were the same as those for the ATc titration experiment, except that the strains were grown for 96 h. Similar to the Scan-o-matic analysis, all downstream analysis was performed using R version 4.0.2.

**Expression analysis by qPCR. (i) Strains.** Expression analysis by qPCR was performed to detect the mRNA expression of *RPN9*, *RPT4*, *GLC7*, and *YPI1*. The strains were selected based on their growth in medium with acetic acid and their importance in the proposed main hypotheses of the study. Rpn9 and Rpt4 are key components of the 19S regulatory particle lid and base complexes, respectively, of the 26S proteosome. CRISPRi targeting of these two genes displayed some of the strongest fitness gains under acetic acid stress. Strains with gRNAs targeting *YPI1* or *GLC7* were selected to add to the literature-based foundation of our hypothesis of Ypi1 and Glc7 interaction. For each target gene, 5 strains (i.e., each with a different gRNA) that showed different degrees of acetic acid tolerance/sensitivity in Scan-o-matic screening were selected. Three control strains (CC2, CC23, and CC32) were included to estimate the expression of the target genes in the absence of CRISPR interference.

**(ii) RNA preparation and cDNA synthesis.** Cells were grown to mid-exponential phase in liquid YNB medium (basal conditions) or YNB medium supplemented with 125 mM acetic acid in the Bioscreen platform and collected by centrifugation at $2,000 \times g$ at 4°C for 3 min. The cell pellet was immediately frozen in liquid nitrogen. Two independent replicates for each CRISPRi strain and three independent replicates for each control strain were included. For RNA preparation, the pellet was dissolved in 600 $\mu$l lysis buffer (PureLink RNA minikit; Invitrogen), after which the cell suspension was transferred into tubes containing 0.5-mm glass beads. Cells were lysed by shaking for 40 s at 6 m/s in an MP Biomedicals FastPrep instrument and then collected by centrifugation in a microcentrifuge for 2 min at 4°C, at full speed. A total of 370 $\mu$l of 70% ethanol was added to the resulting supernatant, and total RNA was prepared using the PureLink RNA minikit (Invitrogen). The obtained RNA was treated with DNase (Turbo DNA-free kit; Invitrogen), and cDNA synthesis was performed on 900 ng DNased RNA using the iScript cDNA synthesis kit (Bio-Rad).

**(iii) Measurement of gene expression.** qPCR was performed using 2.5 ng cDNA and iTaq universal SYBR green supermix (Bio-Rad) for detection. The expression of the target genes was normalized against the geometric mean values of the reference genes *ACT1* and *IPP1*. Primer efficiencies were between 96 and 102% as determined by using different amounts of cDNA. For primer sequences, see Table S1. The qPCR protocol was as follows: an initial denaturation step at 95°C for 3 min, denaturation at 95°C for 20 s, annealing at 60°C for 20 s, and elongation at 72°C for 30 s. In total, 40 PCR cycles were run. For statistical analysis, an *F* test was performed to determine the variance between all the replicates of the control strains and the replicates of a CRISPRi strain. Depending on this result, a two-sample two-tailed *t* test assuming equal or unequal variance was performed for each strain and for a particular condition, where the $H_0$ was m2$^{\Delta CT}$(control) − m2$^{\Delta CT}$(CRISPRi strain) = 0.

**Data availability.** The R scripts used for analysis and the phenomics data generated in this project are available at https://github.com/mukherjeevaskar267/CRISPRi_Screening_AceticAcid. The raw image files of the Scan-o-matic projects can be requested for reanalysis from the authors.

## SUPPLEMENTAL MATERIAL

Supplemental material is available online only.

**FIG S1**, EPS file, 0.3 MB.
**FIG S2**, EPS file, 2.4 MB.
**FIG S3**, EPS file, 0.6 MB.
**FIG S4**, EPS file, 1.6 MB.
**FIG S5**, EPS file, 0.8 MB.
**FIG S6**, EPS file, 0.5 MB.
**FIG S7**, EPS file, 0.6 MB.
**FIG S8**, EPS file, 0.2 MB.
**FIG S9**, EPS file, 0.4 MB.
**TABLE S1**, DOCX file, 0.01 MB.

## ACKNOWLEDGMENTS

We acknowledge the Novo Nordisk Foundation (grant number NF19OC0057685), The Royal Swedish Academy of Sciences, and the Chalmers Area of Advance Energy for financial support.

Y.N. and A.B. conceptualized the project; Y.N., A.B., and V.M. designed the experimental and computational analysis; V.M. and U.L. performed the experiments; V.M. performed computational analysis; V.M., Y.N., A.B., and R.P.S.O. interpreted the results; V.M. and Y.N. wrote the initial draft of the paper; and all authors revised the initial draft and wrote the final paper.

R.P.S.O. is a cofounder of Recombia Biosciences, which engineers yeast to improve industrial fermentation processes.

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
