## [Reviewer comments · mSystems]

A CRISPRi screen of essential genes reveals that proteasome regulation dictates acetic acid tolerance in *Saccharomyces cerevisiae*

Vaskar Mukherjee, Ulrika Lind, Robert St. Onge, Anders Blomberg, and Yvonne Nygård

Corresponding Author(s): Yvonne Nygård, Chalmers University of Technology

Review Timeline:

Submission Date:	April 5, 2021
Editorial Decision:	May 19, 2021
Revision Received:	June 29, 2021
Accepted:	July 8, 2021

Editor: Claudia Vickers

Reviewer(s): Disclosure of reviewer identity is with reference to reviewer comments included in decision letter(s). The following individuals involved in review of your submission have agreed to reveal their identity: Benjamin Luke Schulz (Reviewer #1)

Transaction Report:

DOI: <https://doi.org/10.1128/mSystems.00418-21>

May 19, 2021

Dr. Yvonne Nygård
Chalmers University of Technology
Department of Biology and Biotechnology
Gothenburg
Sweden

Re: mSystems00418-21 (A CRISPRi screen of essential genes reveals that proteasome regulation dictates acetic acid tolerance in *Saccharomyces cerevisiae*)

Dear Dr. Yvonne Nygård:

Thank you for submitting your manuscript to mSystems. We have completed our review and I am pleased to inform you that, in principle, we expect to accept it for publication in mSystems. However, acceptance will not be final until you have adequately addressed the reviewer comments.

It appears that this submission has >10 supplemental items included. Please reduce the number of supplements in your revised submission.

Thank you for the privilege of reviewing your work. Below you will find instructions from the mSystemseitorial office and comments generated during the review.

Preparing Revision Guidelines

For complete guidelines on revision requirements, please see the Instructions to Authors at <https://msystems.asm.org/sites/default/files/additional-assets/mSys-ITA.pdf>. **Submissions of a paper that does not conform to mSystems guidelines will delay acceptance of your manuscript.**

Sincerely,

Claudia Vickers

Editor, mSystems

Journals Department
Reviewer comments:

Reviewer #1 (Comments for the Author):

In this study, Mukherjee et al search for genetic mediators of resistance or sensitivity to acetic acid using a CRISPRi screen of essential genes in yeast. They use the Scan-o-matic platform to perform the initial screen, assaying growth rates on solid media, and then validate select hits using growth in liquid media. As the screen is based on relative growth rates in the absence and presence of acetic acid, it is able to identify genes whose repression is associated with either sensitivity or resistance to acetic acid. GO term enrichment analysis identified sensitivity to acetic acid being associated with repression of genes involved in organelle transport, and resistance to acetic acid being associated with repression of genes involved in the 19S regulatory particle of the 26S proteasome. The authors present a model suggesting decreased abundance of the 19S regulatory particle increases the abundance of the 20S core particle to allow more efficient ATP-independent degradation of oxidized protein. This model is supported by literature, but no additional biochemical validation has been performed in this study.

The screen and targeted validation are technically impressive and convincing. I have some specific comments and questions below.

This study screened for essential genes whose repression could change acetic acid sensitivity or resistance. Repression of essential genes is likely to inhibit growth, which may be important in the use of such a system in industrial applications. The authors could perhaps comment on the absolute effects on growth or viability of the gRNAs that are most promising for providing acetic acid resistance.

The screen for acetic acid tolerance was performed at 150 mM acetic acid, quite a high concentration. What acetic acid concentrations are encountered under growth on lignocellulosic-derived biomass? Would repression of the selected genes also provide acetic acid resistance at lower concentrations?

On what basis were the set of strains chosen for qPCR measurement of transcriptional repression (Fig. 4)?

The subheading "Adapting proteasomal degradation of oxidized proteins to save ATP increases acetic acid tolerance" implies that this mechanism has been biochemically validated. Although the model the authors present is consistent with their results and with the literature, this subheading could be re-worded to be slightly less mechanistic.

The Materials and Methods do provide full details of the number and identity of the genes included in the screen, as well as the statistical analyses. However, these important elements of experimental design and analysis are only very briefly described in the main text - it would improve clarity if these were summarised at appropriate places in the Results.

Essential genes have also been studied systematically in yeast using the tetracycline-regulatable repressor (TetR) system to directly repress target genes. This should be briefly mentioned, perhaps in the discussion.

I have also posted this review as a public comment on the bioRxiv preprint (<https://www.biorxiv.org/content/10.1101/2021.04.06.438748v1>).

Congratulations on an interesting study,
Benjamin L. Schulz, The University of Queensland.

Reviewer #2 (Comments for the Author):

In this paper, Mukherjee and co-authors describe a CRISPRi knockdown screen in yeast to identify essential yeast genes that increase tolerance or sensitivity to acetic acid. The authors screen over 9000 different knockdown strains to identify ~1000 strains whose fitness is affected by acetic acid, and validated 183 hits in a liquid growth assay. The effects of 4 hit-genes were further validated by qPCR. The authors extensively discuss the results of their screen, and identify a role for Glc7 and components of the proteasome regulatory particle in acetic acid toxicity.

Overall, the paper is well-written and the results are clearly described. All data are included in the supplement. However, I have a few concerns, outlined below, about the screen quality and the interpretation of the results.

Main comments:

1. On page 6, the authors say "The CRISPRi strains showed limited phenotypic effects in basal condition and the generation time of 8958 strains (99% of the strains in the library) was within 10% of the generation time of the control strain". This is surprising to me given that they target mainly essential genes. I assume that when the authors say "basal condition", they mean with CRISPRi induction but without acetic acid. I understand that not every gRNA will be effective in a CRISPRi experiment, but if less than 1% of the gRNAs that target essential genes give an observable fitness defect, I think this is a point of concern that should be more clearly addressed.

2. Given that only 1% of the gRNAs seems to be effective, I am surprised that the authors find 665 hit genes. In total, their library targets ~1600 genes, which means that ~40% of the targeted genes show up as hits, which seems extremely high, and makes me worry about the robustness of this

screen. Although the authors confirm many of their hits in liquid growth assays, they use the same gRNAs in the screen as in the validation experiments. Thus, it confirms the gRNA has a specific effect, but this could be related to for example gRNA off-target effects. Furthermore, most of the hit genes are identified using only a single gRNA. I think confirmation of some of the hits using for example temperature-sensitive or other types of mutants, or at least different gRNAs than those that were used in the screen, is needed to prove robustness of the results.

3. On several places in the manuscript the authors describe previous studies that screened mutant libraries for altered acetic acid sensitivity, especially in places where the authors' findings are similar to those of previous screens. However, I do not get a clear overview of the overlap between the current screen and previous screens. Where any of the same genes tested? How large was the overlap in hits? Where there any pathways or processes identified previously that were not found in the current screen? And related to this - what is the main argument for performing the current screen, and what is new in their results?

4. The authors identify several subunits of the regulatory particle of the proteasome as hits in their screen. Using liquid growth experiments, they confirm that gRNA targeting regulatory particle lid subunits lead to acetic acid tolerance (Fig 7A). However, the results for the regulatory particle base are a little more dubious, with only one of the gRNAs showing increased tolerance and one showing increased sensitivity (Fig 7A). The authors mention that these contradictory results are caused by different gRNA efficiency, but the supporting data for this is missing. It is not clear how the data in Fig 4B correspond to those in Fig 7A (different number of gRNAs, and which data point in Fig 4B corresponds to which gRNA?). Could the authors either show qPCR data for the gRNAs using in Fig 7A, or growth of the strains used in Fig 7A in absence of acetic acid?

Minor comments

5. Page 6-7 - the authors list many numbers on these pages. At different places, they mention 1040, 954, and 498 strains that are sensitive to acetic acid. As far as I understood, some of these numbers are before or after statistical tests or describe different comparisons, but I found this part of the paper confusing. I think just focusing on the strains that show a significant growth difference in acetic acid compared to control media would make this section easier to understand.

6. On page 11, the authors suggest that Ypi1 may negatively regulate Glc7, but do not show that Glc7 is downregulated in Ypi1 knockdown strains. Given that they already have the Glc7 qPCR set up, I think this would be an easy experiment that would make this paragraph a lot more convincing.

Reviewer #1:

This study screened for essential genes whose repression could change acetic acid sensitivity or resistance. Repression of essential genes is likely to inhibit growth, which may be important in the use of such a system in industrial applications. The authors could perhaps comment on the absolute effects on growth or viability of the gRNAs that are most promising for providing acetic acid resistance.

Answer: Our study identified a number of CRISPRi strains, that when grown in medium with acetic acid had a reduced generation time when compared to the control strain. For the most prominent of these strains, the ones targeting the 19S regulatory particle of the proteasome did not show a growth defect. Therefore, we do think that fine-tuning of the expression of some of these proteasomal genes may be an industrially relevant approach for increasing stress tolerance. To clarify this, the following sentence was added to the discussion (lines 435-440):

" Notably, none of the acetic acid tolerant strains with gRNAs targeting the 19S regulatory particle of the proteasome showed a severe growth defect in basal medium (https://github.com/mukherjeevaskar267/CRISPRi_Screening_AceticAcid/blob/main/BIG_DATA/Data2.xlsx), indicating them as potentially good target strains in industrial applications.

The screen for acetic acid tolerance was performed at 150 mM acetic acid, quite a high concentration. What acetic acid concentrations are encountered under growth on lignocellulosic-derived biomass? Would repression of the selected genes also provide acetic acid resistance at lower concentrations?

Answer: The concentrations of acetic acid used in this study are within the range of reported concentrations in lignocellulosic hydrolysates (17 - 250 mM) (reviewed by Ko et al.) and thus realistic in industrial applications. As mentioned in the Introduction, we here try to identify gene targets that when repressed can improve the tolerance to harsh lignocellulosic hydrolysates. However, it might be that some of the strains identified in the screen would not induce a quantifiable fitness gain at lower acetic acid concentrations, which we have not tested. The information on concentrations found in hydrolysates has been added to the introduction, lines 86-89:

.... "and lignocellulosic hydrolysates may contain 1-15 g/l (17-250 mM) acetic acid, depending on the feedstock and pretreatment methods employed (reviewed by Ko al. (18))."

On what basis were the set of strains chosen for qPCR measurement of transcriptional repression (Fig. 4)?

Answer: We regret that the rationale behind the qPCR target selection was not clearly described. We have now included an explanation in the Materials and methods section, lines 726-732:

"The strains were selected based on their growth in medium with acetic acid and their importance in the proposed main hypotheses of the study. Rpn9 and Rpt4 are key components of 19S regulatory particle lid and base complex, respectively, of the 26S proteasome. CRISPRi targeting of these two

genes displayed some of the strongest fitness gains under acetic acid stress. Strains with gRNAs targeting YPI1 or GLC7 were selected to add to the literature-based foundation of our hypothesis of Ypi1 and Glc7 interaction."

The subheading "Adapting proteasomal degradation of oxidized proteins to save ATP increases acetic acid tolerance" implies that this mechanism has been biochemically validated. Although the model the authors present is consistent with their results and with the literature, this subheading could be re-worded to be slightly less mechanistic.

Answer: The subheading was changed to *"Adapting proteasomal degradation of oxidized proteins may save ATP and increase acetic acid tolerance"*

The Materials and Methods do provide full details of the number and identity of the genes included in the screen, as well as the statistical analyses. However, these important elements of experimental design and analysis are only very briefly described in the main text - it would improve clarity if these were summarised at appropriate places in the Results.

Answer: Some additional information was added to the Results section, lines 107-111, added text underlined, to make it easier for the reader to assimilate our experimental design:

*"To identify genes involved in tolerance of *S. cerevisiae* to acetic acid, we performed a high-throughput growth screen of a CRISPRi library (9,078 strains) targeting essential and respiratory growth essential genes with an integrated dCas9-Mxi1 repressor (12). Growth phenotyping of the CRISPRi library was performed using the Scan-o-matic system providing high-resolution growth curves on solid media for each strain (20) (Fig. 1). The potential bias between plates and runs was minimized via spatial normalization over the plates. The screens were independently duplicated, in total resulting in >27,000 images, and the image analysis generated >42 million data-points and >140,000 growth curves. Our large-scale screen showed rather good repeatability (Fig. 2A)."*

Essential genes have also been studied systematically in yeast using the tetracycline-regulatable repressor (TetR) system to directly repress target genes. This should be briefly mentioned, perhaps in the discussion.

Answer: Indeed, for a subset of essential genes, gene-by-phenotype analyses have been performed in yeast through TetR-based promoter engineering. We now mention this in the discussion, lines 328-331 (the references for this were added to the reference list):

"For a sub-set of essential genes in yeast, systematic studies have been reported by employing a tetracycline-regulatable repressor (TetR) system-based promoter engineering (3, 32, 33), but this study represents the first example where >99% of the essential genes were individually screened".

Reviewer #2:

1. On page 6, the authors say "The CRISPRi strains showed limited phenotypic effects in basal condition and the generation time of 8958 strains (99% of the strains in the library) was within 10% of the generation time of the control strain". This is surprising to me given that they target mainly essential genes. I assume that when the authors say "basal condition", they mean with CRISPRi induction but without acetic acid. I understand that not every gRNA will be effective in a CRISPRi experiment, but if less than 1% of the gRNAs that target essential genes give an observable fitness defect, I think this is a point of concern that should be more clearly addressed.

Answer: When we compared the qPCR data of the selected strains to their growth phenotype in basal condition, we observed that although the CRISPRi system induced a large range of repression of the target gene by different gRNAs, the phenotype remained largely stable in the basal condition (without acetic acid). This data is now included in Fig. S3. It may be that, in basal medium that provides an optimal environment for the cells to grow, the repression of most of the essential genes does not give a fitness defect until a critical protein level threshold is reached, as cells have developed robustness against changing the amount of proteins. The repression of our CRISPRi strains is initiated at the start of the screen (i.e. preculture is without the gRNA-inducer ATc) and in basal medium the strains grow rapidly under a limited number of generations, which may explain why the CRISPRi based repression system did not reach the critical protein threshold that is necessary to induce fitness defect in basal condition. In fact, this mechanistic explanation was proposed in a recent *E. coli* CRISPRi screen (Donati et al., 2021, Cell Systems 12, 56–67) and we have now added two sentences introducing this theory to the discussion (lines 359-363; the new reference was added to the reference list):

"In basal medium most strains grew rapidly, and it may be that for some strains, the CRISPRi repression that was induced only at the start of the screen did not reach the critical threshold needed to induce growth defects, as was recently shown in an *Escherichia coli* CRISPRi screen, where it was proposed that cells have developed robustness against somewhat changing protein levels (37)."

2. Given that only 1% of the gRNAs seems to be effective, I am surprised that the authors find 665 hit genes. In total, their library targets ~1600 genes, which means that ~40% of the targeted genes show up as hits, which seems extremely high, and makes me worry about the robustness of this screen. Although the authors confirm many of their hits in liquid growth assays, they use the same gRNAs in the screen as in the validation experiments. Thus, it confirms the gRNA has a specific effect, but this could be related to for example gRNA off-target effects. Furthermore, most of the hit genes are identified using only a single gRNA. I think confirmation of some of the hits using for example temperature-sensitive or other types of mutants, or at least different gRNAs than those that were used in the screen, is needed to prove robustness of the results.

Answer: This first part of the comment is in line with the previous point made by the reviewer. We hope our answer above, with the proposed addition to the discussion, will have convinced the reviewer that the lack of phenotypes upon repression of essential genes of strains grown in basal medium were not due to most gRNAs not working but may rather be due to the cells ability to initially counteract changes in protein levels. After longer incubations/growth the protein levels will decrease sufficiently to reveal phenotypic penetrance from the repression (like in the case of acetic acid medium).

The reviewer writes “Given that only 1% of the gRNAs seems to be effective...”. That is incorrect. 1% of the strains did not grow in basal condition, which does not mean that only 1% of gRNA effectively induced repression. Our qPCR results for the selected strains clearly suggest that most gRNAs induced different degrees of repression of the target genes, but with our experimental design resulted in no significant phenotypic change in growth in the basal condition (data on growth now added to Fig. S3). As we validated our findings in a separate liquid growth experiment under two different acetic acid concentrations (125mM and 150mM) and with three independent replicates each, we have high confidence in the results of the screening in acetic acid. As with all studies, strain or set-up dependent differences may lead to some specific phenotypes not being possible to reproduced in other set-ups.

The reviewer writes that “most of the hit genes are identified using only a single gRNA”. We agree that many statistically significant hits were the result of CRISPRi targeting by a single gRNA. Since the gRNAs induce different degree of target gene repression and many gRNAs are known not to provide a phenotype (as explained in the discussion, lines 500-504), we expected that only a subset of the gRNAs would induce sufficient repression of a target gene to trigger a fitness gain or defect. Still, our study resulted in plenty of examples where the tolerant (Data 3 at https://github.com/mukherjeevaskar267/CRISPRi_Screening_AceticAcid/blob/main/BIG_DATA/Data3.xlsx) or sensitive (https://github.com/mukherjeevaskar267/CRISPRi_Screening_AceticAcid/blob/main/BIG_DATA/Data4.xlsx) phenotypes were displayed by multiple gRNA. Out of the 370 tolerant genes, 82 (22%) showed fitness with at-least 2 gRNAs while out of the 367 sensitive genes, 98 (27%) showed fitness with at-least 2 gRNAs.

Finally, screening temperature sensitive mutants on itself is a big undertaking as each strain needs to be phenotyped at a different temperature optimal. Moreover, as mentioned in the discussion (lines 319-321), there will be temperature-dependent side effects (interaction between temperature and acetic acid; acetic acid inhibition has been demonstrated to be much more sever at elevated temperatures (Pitno et al., 1989, Biotechnol Bioeng, 33: 1350-1352) that will make it very tricky to provide conclusive results. Therefore, we decided to leave this outside the scope of this project. Similarly, as strong overexpression of essential genes is often seen to be detrimental, screening overexpression mutants available was not prioritized.

3. On several places in the manuscript the authors describe previous studies that screened mutant libraries for altered acetic acid sensitivity, especially in places where the authors' findings are similar to those of previous screens. However, I do not get a clear overview of the overlap between the current screen and previous screens. Where any of the same genes tested? How large was the overlap in hits? Where there any pathways or processes identified previously that were not found in the current screen? And related to this - what is the main argument for performing the current screen, and what is new in their results?

Answer: There is little overlap between the current screen and the previously published screens since no screens where the involvement of essential genes in acetic acid sensitivity have been reported. Nonetheless, the respiratory growth-essential genes (505 genes) are also included in the EUROSCARF deletion collection that was screened for tolerance towards acetic acid in a few different settings (Kawahata et al., 2006, FEMS Yeast Res 6:924-936; Mira et al., 2010, Microb Cell Fact 9:79; Sousa et al., 2013, BMC Genomic 14:838), the overlap between these studies already

being only minor, in terms of phenotypes of a specific mutant. Nonetheless, when we performed GO enrichment analysis of genes identified to alter acetic acid sensitivity, there was a clear overlap to GO terms enriched in the screens done with the deletion collection. For clarity, we have revised the text referring to this, see lines 255-261:

“We conclude that organelles and vesicle transport were highly enriched among sensitive strains, much in line with findings of earlier deletion collection screens, identifying that these features are important for normal growth in acetic acid (21, 22). Still, the overlap between the deletion strain collection is limited to the respiratory growth essential genes, resulting in low overall overlap between earlier findings, highlighting the need and novelty of our screen with essential genes.”

For us it is clear that screening a library of strains where essential genes are repressed is very novel as this has not been done previously for acetic acid or other stressors. In the discussion (lines 322-328, 334-340) we mention other systematic gene-by-phenotype analyses of essential genes and the limitations of those methods, highlighting the advantages of our methodology. We have throughout the text and in the abstract clearly stated that the library screened targets essential and respiratory growth-essential genes and conclude with describing that

“fine-tuning of the expression of proteasomal genes leading to increased tolerance to acetic acid suggests that this could be a novel strategy for increasing stress tolerance.”

With the addition made at the beginning of the discussion (lines 328-331) we hope that we now convince the reader, and the reviewer, that our approach is novel:

“For a sub-set) of essential genes, systematic studies have been reported by employing a tetracycline-regulatable repressor (TetR) system-based promoter engineering in S. cerevisiae (3, 32, 33), but this study represents the first example where >99% of the essential genes were individually screened.”

4. The authors identify several subunits of the regulatory particle of the proteasome as hits in their screen. Using liquid growth experiments, they confirm that gRNA targeting regulatory particle lid subunits lead to acetic acid tolerance (Fig 7A). However, the results for the regulatory particle base are a little more dubious, with only one of the gRNAs showing increased tolerance and one showing increased sensitivity (Fig 7A). The authors mention that these contradictory results are caused by different gRNA efficiency, but the supporting data for this is missing. It is not clear how the data in Fig 4B correspond to those in Fig 7A (different number of gRNAs, and which data point in Fig 4B corresponds to which gRNA?). Could the authors either show qPCR data for the gRNAs using in Fig 7A, or growth of the strains used in Fig 7A in absence of acetic acid?

Answer: We added the gRNA name as label to each of the datapoint in Fig. 4, making it easier for the reader to correlate the expression level to the phenotypes depicted in Fig 7. We have qPCR data of five strains targeting *RPT4*, as shown in Fig. 4B and Fig. S3B (previously Fig S4B). As suggested, we added the data on the generation time of the strains tested by qPCR to Fig. S3.

We want to point out that in Fig. 7A, the RPT4-TRg-1, the gRNA that induced the highest acetic acid sensitivity, also induced the highest repression of *RPT4*. We believe that since *RPT4* is an essential gene, expression below a critical threshold is overwhelming the physiological system of the strain.

Therefore, all fitness benefits against acetic acid stress observed in mild repression level, disappeared when *RPT4* was highly repressed. This was discussed in lines 497-500:

“Nonetheless, the generation time of RPT4-TRg-1 with a clear repression of RPT4 was increased. We argue that a too strong repression of an essential gene is likely to be detrimental, highlighting the need for a fine-tuned expression when engineering tolerance.”

5. Page 6-7 - the authors list many numbers on these pages. At different places, they mention 1040, 954, and 498 strains that are sensitive to acetic acid. As far as I understood, some of these numbers are before or after statistical tests or describe different comparisons, but I found this part of the paper confusing. I think just focusing on the strains that show a significant growth difference in acetic acid compared to control media would make this section easier to understand.

Answer: We agree with the reviewer that the following part was redundant and may cause some confusion: “Eleven % of all the strains (i.e. 954 strains, including 108 strains that did not grow in acetic acid) in the library had an increased relative generation time, while 19% of all the strains (1,704) had a decreased relative generation under acetic acid stress (Fig. 2C)”, lines 141-144 of original manuscript). This sentence is now removed from the manuscript.

6. On page 11, the authors suggest that Ypi1 may negatively regulate Glc7, but do not show that Glc7 is downregulated in Ypi1 knockdown strains. Given that they already have the Glc7 qPCR set up, I think this would be an easy experiment that would make this paragraph a lot more convincing.

Answer: The negative regulation we referred to is based on literature and was expected to be at the protein level, as stated in lines 265-267: “*GLC7* encodes a type 1 protein phosphatase that contributes to the dephosphorylation and hence activation of glycogen synthases (25).” Therefore, we did not expect a change in the transcriptional level of *GLC7* in strains where *YPI1* is repressed. However, as the reviewer requested this, we now performed the suggested experiment, but expectedly found no correlation between *YPI1* repression and *GLC7* expression levels. The following sentences have now been included to emphasize that the regulation of Glc7 by Ypi1 is presumably acting at the protein level (lines 278-281):

“We tested if the effect of repression of YPI1 was reflected in a change in GLC7 transcripts by qPCR analysis (data not shown). However, we scored no impact on GLC7 mRNA levels, indicating that the potential regulation of Glc7 by Ypi1 would be at the protein level.”

July 8, 2021

Dr. Yvonne Nygård
Chalmers University of Technology
Department of Biology and Biotechnology
Gothenburg
Sweden

Re: mSystems00418-21R1 (A CRISPRi screen of essential genes reveals that proteasome regulation dictates acetic acid tolerance in *Saccharomyces cerevisiae*)

Dear Dr. Yvonne Nygård:

Your manuscript has been accepted, and I am forwarding it to the ASM Journals Department for publication. For your reference, ASM Journals' address is given below. Before it can be scheduled for publication, your manuscript will be checked by the mSystems senior production editor, Ellie Ghatineh, to make sure that all elements meet the technical requirements for publication. She will contact you if anything needs to be revised before copyediting and production can begin. Otherwise, you will be notified when your proofs are ready to be viewed.

As an open-access publication, mSystems receives no financial support from paid subscriptions and depends on authors' prompt payment of publication fees as soon as their articles are accepted. =

Publication Fees:
